# Beyond Imitation: Learning Key Reasoning Steps from Dual Chain-of-Thoughts in Reasoning Distillation

## Abstract

As Large Language Models (LLMs) scale up and gain powerful Chain-of-Thoughts (CoTs) reasoning abilities, practical resource constraints drive efforts to distill these capabilities into more compact Smaller Language Models (SLMs). We find that CoTs consist mainly of simple reasoning forms, with a small proportion ($\approx 4.7\%$) of key reasoning steps that truly impact conclusions. However, previous distillation methods typically involve supervised fine-tuning student SLMs only on correct CoTs data produced by teacher LLMs, resulting in students struggling to learn the key reasoning steps, instead imitating the teacher's reasoning forms and making errors or omissions on these steps. To address these issues, drawing an analogy to human learning, where analyzing mistakes according to correct solutions often reveals the crucial steps leading to successes or failures, we propose mistak**E**-**D**riven key reason**I**ng step distilla**T**ion (**EDIT**), a novel method that further aids SLMs learning key reasoning steps rather than mere simple fine-tuning. Firstly, to expose these crucial steps in CoTs, we design specific prompts to generate dual CoTs data with similar reasoning paths but divergent conclusions. Then, we apply the minimum edit distance algorithm on the dual CoTs data to locate these key steps and optimize the likelihood of these steps. Extensive experiments validate the effectiveness of EDIT across both in-domain and out-of-domain benchmark reasoning datasets. Further analysis shows that EDIT can generate high-quality CoTs with more correct key reasoning steps. Notably, we also explore how different mistake patterns affect performance and find that EDIT benefits more from logical errors than from knowledge or mathematical calculation errors in dual CoTs[1].

## 1 Introduction

With the rapid growth in model size and pre-training data, LLMs have demonstrated impressive CoT reasoning performance in natural language processing (NLP) (Brown et al., 2020; Hoffmann et al., 2022; Chowdhery et al., 2023; OpenAI, 2023b). However, due to the giant model architecture and massive parameters (e.g. GPT-3 (Brown et al., 2020) with 175 billion parameters), the deployment of LLMs in resource-constrained environments becomes challenging.

To address this, researchers (Xu et al., 2023; Jiang et al., 2023b) have explored distilling knowledge from LLMs into smaller language models (SLMs) via instruction-tuning, as seen in LMs like Alpaca (Taori et al., 2023) and Vicuna (Chiang et al., 2023). Despite progress, these distilled models often struggle with complex causal reasoning. To enhance this capability, some studies (Magister et al., 2023; Ho et al., 2023; Fu et al., 2023; Chen et al., 2024; Zhou & Ai, 2024) explore distilling the CoT reasoning ability from LLMs of over 100B parameters (Wei et al., 2022a;b) by fine-tuning on CoTs data annotated by teacher LLMs, known as standard CoTs distillation. Besides, other studies (Hsieh et al., 2023; Li et al., 2022; Liu et al., 2023) propose distilling CoTs within a multi-task learning framework by incorporating additional objectives. However, CoTs usually consist mainly of simple reasoning forms, with a small proportion ($\approx 4.7\%$[2]) of key reasoning steps that are pivotal moments

---

[1]Code can be found at `https://anonymous.4open.science/r/eb77sh-F564`

[2]We calculated the edit distance and its average proportion in the overall sequence on the dual CoT dataset mentioned in our subsequent methods §3.

Figure 1: Examples of CoTs generated by teacher LLMs and student SLMs on our test dataset. Simply SFT leads to an "unthinking" student who imitates the teacher's reasoning forms but makes errors and omissions in key reasoning steps, where the imitated contents are highlighted in red, and the key steps are marked with boxes .

in reasoning that significantly influence subsequent thought processes and conclusions. The essence of the above methods is the simple Supervised Fine-Tuning (SFT) paradigm, where the student model is trained solely on the teacher's correct reasoning data. This paradigm may result in students **struggling to learn the key reasoning steps, instead imitating the teacher's reasoning forms and making errors or omissions on these steps**, as illustrated in Figure 1. Drawing an analogy to human learning, where analyzing mistakes according to correct solutions often reveals the key reasoning steps leading to successes or failures, we propose a novel mistak**E**-**D**riven key reason**I**ng step distilla**T**ion (**EDIT**). This approach focuses on dual CoTs data, encompassing both positive and negative examples of teachers' reasoning. By examining dual CoTs, students can identify and learn from the crucial reasoning steps, thereby improving their CoTs. Specifically, we first retain all CoTs data annotated by the teacher, irrespective of correctness. Subsequently, we design two comprehensive prompts to instruct teachers to produce dual CoTs that share similar intermediate reasoning steps but lead to divergent conclusions. Finally, we utilize the minimum edit distance algorithm to locate key reasoning steps in dual CoTs, as shown in Figure 3, and then utilize a fine-grained loss function to optimize the likelihood of these steps.

Extensive experiments show that the student models distilled by EDIT exhibits higher performance and generalization than the baselines on both in-domain (IND) and out-of-domain (OOD) benchmark reasoning datasets. Further analyses indicate that EDIT can generate higher-quality CoTs with more correct key reasoning steps by auto evaluation and case studies. Notably, we also show EDIT can benefit more from logical mistake patterns than knowledge or mathematical calculation errors in dual CoTs, potentially paving the way for future research on the efficient use of mistakes.

Our contributions can be summarized as follows:

1. We reveal a shortfall in the previous distillation methods, where the simple SFT paradigm may result in students mimicking the teacher's reasoning forms but making errors or omissions in key reasoning steps, thus diminishing the versatility of CoTs.

2. We propose mistake-driven key reasoning step distillation, which allows students to learn key reasoning steps from our specifically designed dual CoTs data, further improving reasoning.

3. Extensive experiments validate the effectiveness of our method across both IND and OOD datasets, showing that EDIT can reduce errors in key reasoning steps for students.

4. We investigate how different mistake patterns impact EDIT and find that logical errors provide the more significant benefits than knowledge or mathematical calculation errors.

## 2 RELATED WORKS

**CoT Reasoning.** The emergent ability appears in LLMs across a wide range of NLP tasks (Chowdhery et al., 2023; Wei et al., 2022a). One such ability is CoT reasoning, which involves generating a

series of intermediate reasoning steps. This ability has been further explored recently with the release of OpenAI's o1 model (OpenAI, 2024). While CoT prompting techniques (Wei et al., 2022b) significantly enhance the problem-solving capabilities of models (Kojima et al., 2022; Wang et al., 2023b; Huang et al., 2023), it has little effect on smaller models (Wei et al., 2022a). Chung et al. (2022) suggest that CoT reasoning can be induced in SLMs via instruction tuning on CoTs data. Our work show that the CoT capabilities of SLMs can be further improved by learning from key reasoning steps in dual CoTs data.

**Knowledge Distillation from LLMs.** There has been a lot of work dedicated to distilling knowledge (Hinton et al., 2015) from powerful proprietary LLMs, e.g. ChatGPT (OpenAI, 2023a) in a black-box setting. However, most of these works primarily focus on the general ability distillation by instruction tuning on large and diverse datasets (Peng et al., 2023; Jiang et al., 2023b; Li et al., 2024). In contrast, we aim to distill the CoT reasoning capabilities from LLMs same as the standard CoTs distillation (Magister et al., 2023; Ho et al., 2023). Besides, some studies (Li et al., 2022; Hsieh et al., 2023; Liu et al., 2023) employ LLM's rationale or self-evaluation output to enhance SLM's reasoning in a multi-task learning framework. Fu et al. (2023) fine-tune SLMs on four types of reasoning data to ensure out-of-distribution generalization. Wang et al. (2023c) distill SLMs by learning from self-reflection and feedback from LLMs in an interactive multi-round paradigm. Chen et al. (2024) maximize the mutual information between multi objectives for CoTs distillation. Ranaldi & Freitas (2024) use in-family and out-family teachers to generate more CoTs for fine-tuning students. Different from the above works, we assist CoTs distillation with teachers' mistakes to alleviate the style imitation of teachers' reasoning.

**Learning from Mistakes.** Recent studies use mistake data to enhance the performance of LMs. Shinn et al. (2023) propose Reflexion that allows the LLM agent to self-reflect from its mistakes. Wang & Li (2023) introduce a study assistant that collects and retrieves LLMs' training mistakes to guide future inferences. Li et al. (2023) propose CoK that corrects potential mistakes in the rationale by retrieving knowledge to avoid error propagation. However, both of the above methods require the models to be large enough to have basic CoT reasoning or instruction-following capabilities, which is almost impossible to occur in vanilla SLMs. Wang et al. (2023a) propose fine-tuning on counterfactual data to ensure the faithful reasoning of the student model. An et al. (2023) propose LEMA that fine-tunes language models on corrected mistake data, where the mistakes are collected from various LLMs e.g. LLaMA2-70B (Touvron et al., 2023), WizardLM-70B (Xu et al., 2023), and corrected by GPT-4 (OpenAI, 2023b). Additionally, Sun et al. (2024) propose Retrieved In-Context Principles, which retrieve mistakes to provide customized guidance and improve model performance during inference. In contrast, we collect the teachers' mistakes to create a dual CoTs dataset for further key reasoning steps learning on model distillation.

## 3 MISTAKE-DRIVEN KEY REASONING STEP DISTILLATION

We present the overview of our proposed method in Figure 2. Concretely, (1) unlike prior works (Magister et al., 2023; Hsieh et al., 2023) that only focus on correct CoTs annotated by teacher LLMs, we first retain all CoTs reasoning data, regardless of its correctness. (2) Then based on the previously retained correct and wrong CoTs, we construct dual CoTs datasets consisting of positive-negative CoT pairs that follow similar intermediate reasoning steps but lead to divergent conclusions. Specifically, we design two comprehensive contextual prompts to instruct teacher LLMs to rectify the originally wrong CoTs and corrupt originally correct CoTs. (3) Finally, we distill the student SLMs by training on the teacher's correct CoTs reasoning data and further Key Reasoning Steps Learning (KRSL) on the dual CoTs datasets.

### 3.1 COTS ANNOTATED BY LLMS

We utilize CoT Prompting (Wei et al., 2022b) to extract CoTs for a raw dataset $\mathcal{D} = \{(q, a)\}$ from LLMs, where $q$ is the question and $a$ is the golden answer. Specifically, we first create a CoTs Extraction Prompt CEP that contains several human-curated question-CoTs pair examples and the task description, which can be found in Appendix C.1. For each $q \in \mathcal{D}$, we extract CoTs as:

$$CoT \sim LLM\left(\text{CEP} \oplus q\right) \tag{1}$$

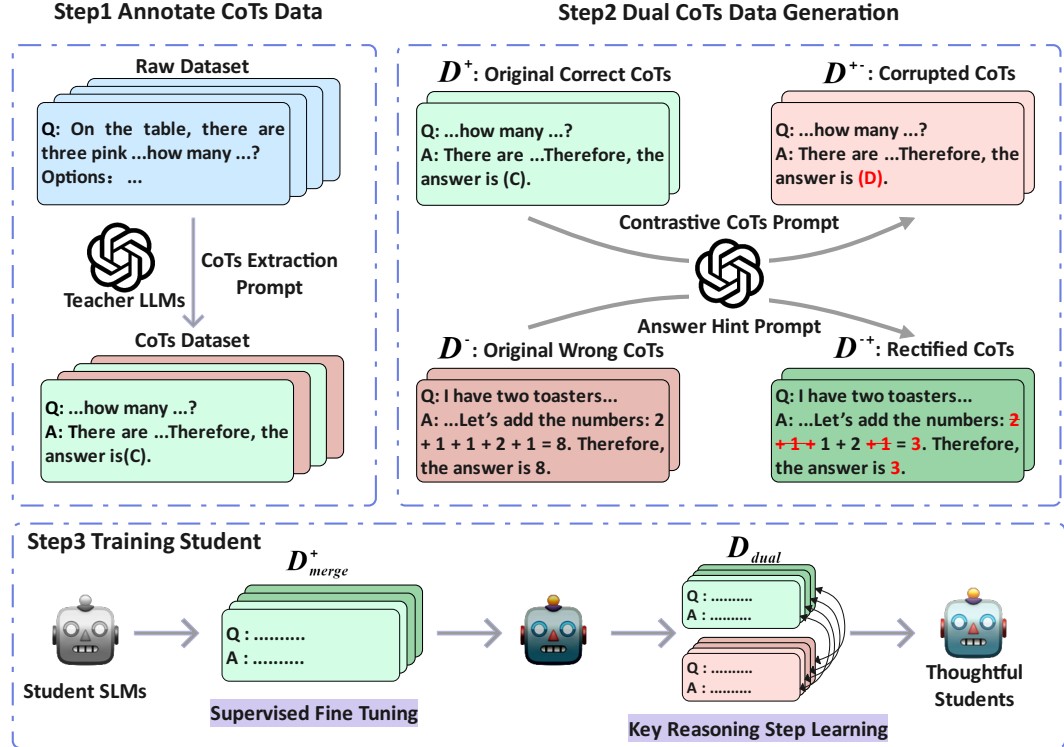

Figure 2: **Overview of our mistake-driven key reasoning step distillation.** (1) We first retain all CoTs data annotated by teacher LLMs (2) and ask teacher LLMs to generate dual CoTs data using our designed two comprehensive prompts. (3) Then we fine-tune student SLMs on both original correct and rectified-after CoTs data. Finally, we apply key reasoning step learning on the pre-tuned student SLMs by identifying the minor difference between the dual CoTs.

where $\oplus$ means concatenation. Then, we classify the CoTs annotated dataset into two datasets according to the final answer's correctness[3], same as Zelikman et al. (2022). One is the CoTs-original correct dataset $\mathcal{D}^+ = \{(q, CoT^+) \mid \forall (q, a) \in \mathcal{D}, \hat{a} = a \ \& \ \hat{a} \in CoT^+\}$ and the other is CoTs-original wrong dataset $\mathcal{D}^- = \{(q, CoT^-) \mid \forall (q, a) \in \mathcal{D}, \hat{a} \neq a \ \& \ \hat{a} \in CoT^-\}$.

## 3.2 DUAL CoTs GENERATION

We define dual CoTs data as contrasting CoTs that follow similar reasoning steps but reach divergent conclusions compared to the original. To provide a deeper understanding, we also present several examples of dual CoTs in Appendix A. In the following, we will introduce how to generate dual CoTs datasets including $\mathcal{D}^{+-}$ contrasting to $\mathcal{D}^+$, and $\mathcal{D}^{-+}$ contrasting to $\mathcal{D}^-$.

**Rectify Wrong CoTs.** To generate correct CoTs contrasting with the originally wrong CoTs, inspired by Rationalization (Zelikman et al., 2022), we design an Answer Hint Prompt `AHP` that shares the same examples with `CEP` but with different organizational structures. The template of `AHP` can be found in Appendix C.2. Each example in the context and the final provided question will be inserted with a hint that tells LLMs the answer first before CoTs. Thus, due to the same in-context examples and hint answers, teacher LLM can rectify its original wrong CoTs data with similar reasoning steps but correct answers. For each $q \in \mathcal{D}^-$, we rectify CoTs as follows and then have the Rectified CoTs dataset $\mathcal{D}^{-+} = \{(q, CoT^{-+})\}$:

$$CoT^{-+} \sim LLM\left(\text{AHP} \oplus q \oplus a\right) \tag{2}$$

---

[3]To support our assumption of CoT correctness, We randomly sample 100 examples to manually check the logical consistency between the CoT and the final answer and find that the CoTs generated by ChatGPT generally support the final answer.

**?** On the floor, I see two magenta scrunchiephone chargers, one gold fidget spinner...

**Key Steps in Correct CoTs Data** According to this question, there are a total of 5 scrunchiephone chargers on the floor: 2 magentascrunchiephone chargers and 3 silver scrunchiephone chargers. If we remove all the magenta items from the floor, we remove the 2 magenta scrunchiephone chargers. Therefore, the number of scrunchiephone chargers remaining on the floor is 3. Therefore, the answer is (D).

**Key Steps in Wrong CoTs Data** According to this question, there are two magenta scrunchiephone chargers and three silver scrunchiephone chargers on the floor. If we remove all the magenta items from the floor, we are left with 2 silver scrunchiephone chargers. Therefore, the answer is (C).

Figure 3: Examples of locating key reasoning steps in dual CoTs, where the correct CoT and the wrong CoT are dual to each other. The identified key steps in correct reasoning and wrong reasoning are respectively marked in green and red.

**Corrupt Correct CoTs.** To generate incorrect CoTs contrasting with the originally correct CoTs, a straightforward approach is to use AHP with incorrect hint answers to prompt LLMs to produce wrong CoTs. However, in practice, we find that LLMs rarely follow the incorrect hints and still generate correct CoTs. This may be due to the simplicity of the questions, which fall within the LLMs' knowledge range. Additionally, LLMs, having undergone Reinforcement Learning from Human Feedback (RLHF) (Ouyang et al., 2022), may resist providing unhelpful answers. Therefore, we design a Contrastive CoTs Prompt (CEP) to entice LLMs to generate incorrect CoTs, leveraging their strong in-context learning capabilities. The prompt template can be found in Appendix C.3. Specifically, to ensure high-quality incorrect CoTs, we randomly sample negative examples from $\mathcal{D}^-$ and positive examples from $\mathcal{D}^{-+}$, pair them, and place them into the CCP as curated joint in-context examples. For each $q \in \mathcal{D}^+$, we corrupt CoTs as follows and then have the corrupted CoTs dataset $\mathcal{D}^{+-} = \{(q, CoT^{+-})\}$:

$$CoT^{+-} \sim LLM\left(\texttt{CCP} \oplus q \oplus CoT^+\right) \tag{3}$$

### 3.3 TRAINING STUDENT WITH COTS

**Surpervised Fine-tuning on Correct CoTs.** After preparing the dual CoTs, we first fine-tune student models on the teachers' original correct CoTs dataset $\mathcal{D}^+$ and rectified CoTs dataset $\mathcal{D}^{-+}$. The training objective is as follows:

$$\pi_{sft} = \arg\max_\pi \mathbb{E}_{q,CoT \sim \mathcal{D}^+_{merge}}\left[\log \pi(CoT \mid q)\right] \tag{4}$$

where the merged correct CoTs dataset $\mathcal{D}^+_{merge} = \mathcal{D}^+ \cup \mathcal{D}^{-+}$, and $\pi_{sft}$ denotes the student with the base inference ability after the initial fine-tuning.

**Key Reasoning Steps Learning** Inspired by (Guo et al., 2023b) who leverage fine-grained quality signals to align human preference, we propose a key reasoning steps learning (KRSL) method to further encourage students to comprehend the reasons behind both correct and wrong CoTs.

*Step1.* We pair the teacher's original correct CoTs dataset $\mathcal{D}^+$ with its corrupted CoTs dataset $\mathcal{D}^{+-}$, creating an originally correct dual CoTs dataset $\mathcal{D}^+_{dual} = \{(q, CoT^+, CoT^{+-})\}$, where $CoT^+$ and $CoT^{+-}$ are dual to each other; similarly, the teacher's inherently wrong dual CoTs dataset $\mathcal{D}^-_{dual} = \{(q, CoT^{-+}, CoT^-)\}$. By merging them, we obtain the ultimate dual CoTs datasets $\mathcal{D}_{dual} = \mathcal{D}^+_{dual} \cup \mathcal{D}^-_{dual}$, which is prepared for the subsequent learning of key reasoning steps.

*Step2.* Then we employ the minimum edit distance to identify the key steps in both correct reasoning and wrong reasoning, as shown in Figure 3. In this way, students can identify less frequent text segments that are inserted or replaced in wrong CoTs compared to correct CoTs, and vice versa. These text segments are considered key reasoning steps. After that, we assign token-level weights to facilitate fine-grained learning for correct CoTs and wrong CoTs in $D_{dual}$ respectively:

$$\omega_t^+ = \begin{cases} \alpha, & \text{if } CoT_t^+ \text{ is inserted or replaced} \\ 0, & \text{otherwise} \end{cases}, \omega_t^- = \begin{cases} \beta, & \text{if } CoT_t^- \text{ is deleted or replaced} \\ 0, & \text{otherwise} \end{cases}. \tag{5}$$

where $\alpha \geq 0$, $\beta \geq 0$ and $\omega_t^+$ represents the weight of $t$-th token in the correct CoTs (semantically same with $\omega_t^-$). We set the weights to zero to ignore the impact of identical tokens in the dual CoTs.

***Step3.*** Finally, to ensure that the student makes correct decisions on key steps in correct reasoning, we optimize the student model on these tokens with weighted negative log-likelihood. Conversely, to prevent the student from making key steps present in wrong reasoning, we optimize the student model on these steps with weighted positive log-likelihood. The sum of both is taken as the final loss. The optimization objective is as follows:

$$\max_{\pi_{sft}} \mathbb{E}_{q,CoT^+,CoT^- \sim \mathcal{D}_{dual}} \left[ \mathcal{L}(\pi_{sft}, q, CoT^+, \omega^+) - \mathcal{L}(\pi_{sft}, q, CoT^-, \omega^-) \right] \quad (6)$$

where

$$\mathcal{L}(\pi, q, CoT, \omega) = - \sum_{CoT_t \in CoT} \omega_t \log \pi(CoT_t \mid q, CoT_{<t}) \quad (7)$$

## 4 EXPERIMENTS

### 4.1 EXPERIMENTAL SETUP

**In-domain (IND) Dataset: BIG-Bench Hard (BBH)**   (Suzgun et al., 2023) consists of 27 challenging tasks that span arithmetic, symbolic reasoning, etc. This collection is mainly composed of multiple-choice questions, along with a minority of open-ended questions. To underscore the superiority of our method, we divide the BBH dataset for each subtask into a training set (BBH-train) for distillation and a test set (BBH-test) for in-domain evaluation, following a 4:1 ratio.

**Out-of-domain (OOD) Dataset: (1) BIG-Bench Sub (BB-sub)**   is derived from the BIG-Bench (BB) (Guo et al., 2023a), which includes 203 tasks covering linguistics, mathematics, common-sense reasoning, etc. To simplify our evaluation, we refine the selection of tasks from BB by identifying those associated with keywords such as "multiple-choice" and "reasoning."[4] Additionally, we exclude any tasks that are part of the BBH dataset, narrowing our pool to 61 distinct subtasks. For each of these subtasks, we randomly sample up to 100 instances, culminating the BB-sub dataset. **(2) AGIEval** (Zhong et al., 2023) is a benchmark that assesses LMs on reasoning capabilities using human exams across various fields, including English, Math, Law, and Logic. We focused on the English multiple-choice questions within this benchmark to evaluate our method's effectiveness. **(3) AI2 Reasoning Challenge (ARC)** (Clark et al., 2018) comprises **ARC-E**asy and **ARC-C**hallenge from middle and high school science exams. ARC-E features simpler questions, while ARC-C includes more challenging ones. We use their test sets for evaluation. Detailed statistics for all mentioned benchmarks are provided in Appendix B.6.1. BigBench, AGIEval, and ARC are standard benchmarks for evaluating LLMs reasoning performance. Specifically, BigBench and AGIEval have been employed in related works (Fu et al., 2023; Jiang et al., 2023b), and ARC is frequently used in technical reports for LLaMA3 (AI@Meta, 2024) and GPT-4 (OpenAI, 2023b).

**Models & Implementation Details.**   We employ the widely-used open-source language model, LLaMA2-7B (Touvron et al., 2023), as our student SLM. For the teacher model, given its performance and cost-effectiveness, we employ OpenAI's advanced black-box LLM, ChatGPT, specifically using the `"gpt-3.5-turbo-0613"` variant for extracting CoTs with the same manual prompt that is used in (Suzgun et al., 2023). We employ LoRA (Hu et al., 2022) for parameter-efficient fine-tuning of the student SLMs. We empirically set $\alpha$ in KRSL as 1.0 and $\beta$ as 0.025. Our experiments leverage a mixed-precision training strategy, carried out on $4 \times$ A100 GPUs. We employ vLLM[5] (Kwon et al., 2023) to enhance inference speed, using a greedy decoding method for text generation on a single A100 GPU. More training details and hyperparameter settings can be found in Appendix B.6.2.

**Baselines.**   We compare EDIT with the following baselines: (1) **Teacher & Vanilla Student** under various settings, e.g., Zero-shot (+ CoT) or Few-shot (+ CoT). (2) **Std-CoT** (Magister et al., 2023), which is a standard CoTs distillation method that directly fine-tunes student SLMs on CoTs data. (3) **MT-CoT** (Li et al., 2022) is a multi-task CoTs distillation strategy that aims to optimize both the prediction of answers and the learning of CoTs concurrently. (4) **SCOTT** (Wang et al., 2023a) aims to bolster the reasoning consistency in the student SLMs by integrating counterfactual data into its training regimen.

---

[4]`https://github.com/google/BIG-bench/blob/main/bigbench/benchmark_tasks/README.md`.

[5]`https://github.com/vllm-project/vllm`

Table 1: Results (Accuracy, %) of the main experiment. w/o RWC represents that student models are distilled without using the rectified teacher's wrong CoTs in the first step of EDIT and w/o KRSL denotes that the second step KRSL in EDIT is removed. The improvements of EDIT and its variants, w/o RWC and w/o KRSL, over the average best baseline are indicated by subscripts. We also provide results of more commonly used reasoning subtasks in Appendix B.1.

| Method | Distill? | BBH-test | BB-sub | AGIEval | ARC-E | ARC-C | AVG |
|---|---|---|---|---|---|---|---|
| In-domain? | | ✓ | ✗ | ✗ | ✗ | ✗ | |
| **Teacher: ChatGPT (gpt-3.5-turbo)** | | | | | | | |
| Zero-shot-CoT | ✗ | 42.7 | 44.1 | 49.5 | 91.9 | 81.1 | 61.9 |
| Few-shot-CoT | ✗ | 73.1 | - | - | - | - | - |
| **Student: LLaMA2-7B** | | | | | | | |
| Zero-shot | ✗ | 14.8 | 15.5 | 6.9 | 18.2 | 13.9 | 13.9 |
| Few-shot | ✗ | 15.1 | 28.5 | 25.5 | 25.5 | 25.4 | 24.0 |
| Zero-shot-CoT | ✗ | 10.6 | 7.7 | 7.1 | 18.4 | 14.8 | 11.7 |
| Few-shot-CoT | ✗ | 16.3 | 25.3 | 9.9 | 17.2 | 17.2 | 17.2 |
| MT-CoT (Li et al., 2022) | ✓ | 56.8 | 30.3 | 22.0 | 49.4 | 38.2 | 39.3 |
| SCOTT (Wang et al., 2023a) | ✓ | 42.4 | 18.8 | 13.0 | 45.7 | 34.1 | 30.8 |
| Std-CoT (Magister et al., 2023) | ✓ | 54.2 | 28.7 | 21.6 | 59.6 | 45.1 | 41.8 |
| EDIT (ours) | ✓ | **60.9**$_{+6.7}$ | **31.1**$_{+2.4}$ | **25.9**$_{+4.3}$ | **64.1**$_{+4.5}$ | **50.5**$_{+5.4}$ | **46.5**$_{+4.7}$ |
| w/o RWC | ✓ | 55.1$_{+0.9}$ | 30.1$_{+1.4}$ | 24.1$_{+2.5}$ | 60.3$_{+0.7}$ | 44.1$_{-1.0}$ | 42.7$_{+0.9}$ |
| w/o KRSL | ✓ | 59.7$_{+5.5}$ | 30.0$_{+1.3}$ | 24.5$_{+2.9}$ | 61.9$_{+2.3}$ | 45.5$_{+0.4}$ | 44.3$_{+2.5}$ |

## 4.2 MAIN RESULTS

We compare EDIT with the baselines across both IND and OOD datasets in Table 1 and illustrate the results by answering the following research questions.

**Can CoT distillation improve the performance of students?** From the table, it is evident that the student SLMs with distillation outperform those that were not distilled. This demonstrates that the reasoning ability of LLMs can be effectively transferred to SLMs by distilling CoTs.

**Can EDIT further enhance the performance of students compared to other distillation methods?** It can be observed that our proposed method EDIT outperforms the distillation baselines on both IND and OOD datasets, achieving an average improvement of 4.7 % compared to the standard CoT distillation (Std-CoT), which demonstrates the effectiveness and generalizability of EDIT.

**How significant are the improvements in EDIT attributed to the rectified wrong CoTs and the key steps learning, respectively?** Ablation results in the table show that removing the rectified wrong CoTs (w/o RWC) and removing key reasoning steps learning (w/o KRSL) result in performance degradation on almost all IND and OOD, emphasizing the importance of both components. On the one hand, the rectified teachers' mistakes aid the students in learning diverse ways of thinking. On the other hand, KRSL directs the student's attention to crucial steps in the dual CoTs, thereby improving the reasoning ability of the students. Additionally, we note that although KRSL and DPO (Rafailov et al., 2023) share very similar learning principles, DPO performed unexpectedly poorly in this scenario. Detailed experiments and analyses are provided in Appendix B.5.

## 4.3 ABLATION STUDY

**EDIT is universally applicable to SLMs with various sizes.** To better adapt to the community's varying computational resource requirements, we conduct experiments on models of different sizes, including TinyLLaMA-1.1B[6] (Zhang et al., 2024), LLaMA2-7B and 13B. The results in Figure 4 show that EDIT outperforms the baselines across different model sizes. Particularly on benchmarks with broader evaluation dimensions such as BB-sub and AGIEval, significant improvements are observed regardless of the model size. This suggests that the more challenging a task is, the more it

---

[6]https://huggingface.co/TinyLlama/TinyLlama-1.1B-intermediate-step-1431k-3T

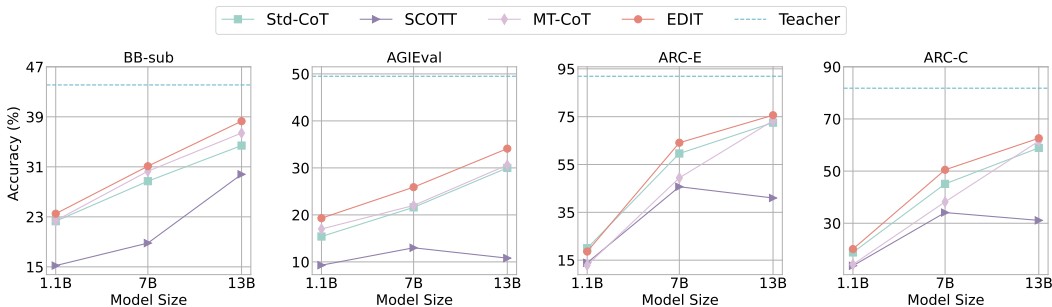

Figure 4: Ablation results on model size for four OOD datasets. The dotted line indicates the performance of the teacher LLM under the Zero-shot-CoT setting. Due to the space limitation, we present the results on the IND dataset in Appendix B.2.

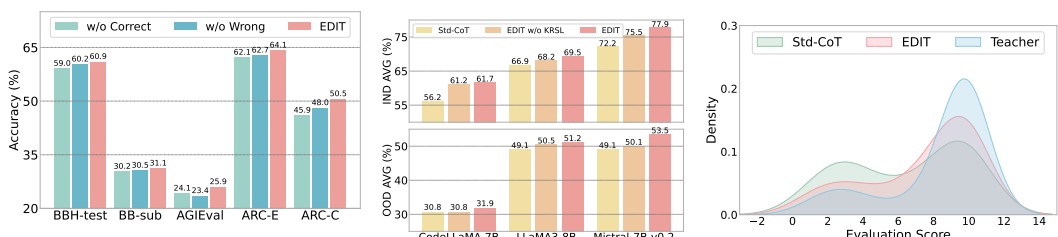

Figure 5: **Left:** Ablation results on key reasoning steps for the IND (BBH-test) and OOD (others) datasets. w/o Correct represents that students only learn key reasoning steps in wrong CoTs and w/o Wrong represents that students only learn key reasoning steps in correct CoTs. **Middle:** Ablation results on different student models for the IND and OOD. We compare EDIT with its variants w/o KRSL and Std-CoT. The results are reported by IND-AVG and OOD-AVG that respectively denote average accuracy on IND and OOD datasets. **Right:** Score distribution evaluated by GPT-4 on BBH-test. We use kernel density estimation to visualize the distribution of CoTs quality scores.

requires genuine reasoning rather than mere imitation, highlighting the benefits that EDIT brings to student SLMs.

**EDIT is universally applicable to SLMs with various architectures.** To cater to the community's diverse model preferences, we conduct experiments on models of different architectures, including CodeLLaMA-7B (Touvron et al., 2023), LLaMA3-8B (AI@Meta, 2024), and Mistral-7B-v0.2 (Jiang et al., 2023a). As shown in Figure 5 (middle), EDIT consistently outperforms its variant w/o KRSL and the baseline Std-CoT across all model architectures. Notably, the performance gap is significantly larger for the stronger model, Mistral, indicating that our method provides greater benefits with more powerful base models.

**Correct key reasoning steps have a greater impact than incorrect ones.** We conduct an ablation study on the key reasoning steps in KRSL where students learn exclusively from either the correct or wrong reasoning steps (referred to §3.3, we set $\alpha = 0$ or $\beta = 0$, respectively). The results shown in Figure 5 (left) indicate that learning key reasoning steps solely from either correct or wrong CoTs leads to a decline in performance. This demonstrates that joint learning from both correct and wrong key reasoning steps is more beneficial for enhancing students' reasoning capabilities. Furthermore, we observe a greater performance drop in the absence of key steps in correct CoTs (w/o Correct) compared to the absence of key steps in wrong CoTs (w/o Wrong), suggesting that key steps from correct CoTs have a more significant impact on students' learning.

**The quality of dual CoTs data is more important than quantity.** We also explore which component of the dual CoTs dataset in KRSL plays a more significant role: the originally correct dual CoTs $\mathcal{D}_{dual}^+$ or the inherently wrong dual CoTs $\mathcal{D}_{dual}^-$. From the Table 2, compared to using $\mathcal{D}_{dual}^+$,

employing $\mathcal{D}_{dual}^-$ resulted in superior performance, even with less data, which demonstrates that $\mathcal{D}_{dual}^-$ has higher data quality compared to $\mathcal{D}_{dual}^-$. The dual CoTs constructed from the inherent wrong CoTs of teachers more effectively highlight the key steps in reasoning.

Table 2: Performance (Accuracy, %) comparison across dual CoTs datasets used in KRSL. The $\mathcal{D}_{dual}^+$ and $\mathcal{D}_{dual}^-$ represents that only the originally correct dual CoTs dataset or the inherently wrong dual CoTs dataset is used in KRSL.

| Dataset | BBH-test | BB-sub | AGIEval | ARC-E | ARC-C | AVG |
|---|---|---|---|---|---|---|
| $\mathcal{D}_{dual}^+$ (# = 3805) | 61.3 | 31.2 | 24.4 | 64.6 | 48.9 | 46.1 |
| $\mathcal{D}_{dual}^-$ (# = 1402) | 60.9 | 30.8 | 26.0 | 63.8 | 50.5 | 46.4 |
| $\mathcal{D}_{dual}$ (# = 5207) | 60.9 | 31.1 | 25.9 | 64.1 | 50.5 | 46.5 |

## 5 ANALYSIS

### 5.1 COST ANALYSIS

Considering that our method utilizes dual CoTs data, which results in twice the amount of training data compared to the baselines, we implement two additional baseline settings to ensure a fair comparison and ablate the impact of the increased data size due to dual CoTs: (1) **Std-CoT w/ Repeat Sampling**. We perform random repeat sampling on the baseline's original training data until the volume matches that of EDIT; (2) **Std-CoT w/ Dual CoTs**. We train the Std-CoT using all data included in EDIT, adding the marker "`[Counterfactual Reasoning]`" before the negative sample's question to differentiate it from positive reasoning. Results in Table 3 show that while Std-CoT benefits from additional data, it underperforms compared to EDIT across most tasks. EDIT's superiority stems from its method of learning key reasoning steps beyond mere imitation, allowing students to learn from mistakes. Additionally, Std-CoT with Dual CoTs outperforms that with Repeat Sampling in OOD tasks by incorporating counterfactual reasoning, reducing overfitting and better generalizing the reasoning. This supports our view that simple fine-tuning with correct teacher data is insufficient for true reasoning learning.

Table 3: Results (Accuracy, %) of the cost analysis.

| Method | Training Data Size | BBH-test | BB-sub | AGIEval | ARC-E | ARC-C | AVG |
|---|---|---|---|---|---|---|---|
| Std-CoT w/ Repeat Sampling | 10414 | 59.4 | 30.3 | 24.0 | 58.0 | 42.1 | 42.8 |
| Std-CoT w/ Dual CoTs | 10414 | 54.8 | **32.9** | 25.1 | 62.2 | 44.1 | 43.8 |
| EDIT (ours) | 10414 | **60.9** | 31.1 | **25.9** | **64.1** | **50.5** | **46.5** |

### 5.2 QUALITY OF GENERATED COTs

Beyond accuracy in reasoning, the quality of CoTs is crucial for interpretable AI. Therefore, we leveraged the sota LLM, GPT-4, to score the quality of CoTs generated by Std-CoT, EDIT, and teacher LLMs. The evaluation focused on which CoT best reflects the key reasoning steps in the problem-solving process, with the prompt template detailed in Appendix C.4. The distribution of the evaluation scores is shown in Figure 5 (right), where we observe that the score distribution for CoTs generated by EDIT is closer to that of the teacher compared to Std-CoT. This illustrates that EDIT is more effective in learning the key reasoning steps, resulting in the production of high-quality CoTs.

### 5.3 CASE STUDY

To more clearly show the quality of key reasoning steps in generated CoTs, we present 5 cases sampled from BBH, AGIEval, and ARC, compared with Std-CoT and teachers, as detailed in Appendix B.3. Tables 19 and 20 show that the reasoning form of the student SLMs distilled by Std-CoT is very similar to that of the teacher. However, the student SLMs distilled by EDIT exhibit a changed way of thinking, leading to the correct answers. Table 21 reveals nearly identical reasoning among the three, yet in the critical reasoning steps 7 and 8, Std-CoT fails to make the correct decisions, whereas EDIT

correctly executes stack operations. Cases from OOD datasets, shown in Tables 22 and 23, indicate that EDIT can accurately analyze problems and provide more logical reasoning.

## 5.4 INTEGRATION WITH SELF-CONSISTENCY

In this subsection, we explore the integration of our method with the widely-used CoT reasoning technique, Self-Consistency (SC). SC improves reasoning performance by generating multiple reasoning paths and selecting the most consistent answer through majority voting. For SC, we apply majority voting with 8 sampled reasoning paths, using `temperature=0.7` and `topp=0.95` for decoding. As shown in Table 4, nearly all CoT distillation methods, including our method EDIT, show significant performance improvements when combined with SC. This demonstrates that EDIT can be effectively integrated with CoT reasoning techniques, providing both flexibility and scalability.

Table 4: Results of Integration with Self-consistency (Accuracy, major vote@8).

| Method + Self-consistency | BBH-test | BB-sub | AGIEval | ARC-E | ARC-C | AVG |
|---|---|---|---|---|---|---|
| MT-CoT | 56.4 | 32.2 | 22.3 | 68.5 | 52.8 | 46.4 |
| SCOTT | 41.1 | 22.0 | 16.7 | 56.1 | 40.6 | 35.5 |
| Std-CoT | 56.3 | 31.2 | 25.2 | 66.2 | 50.0 | 45.8 |
| Std-CoT w/ Repeat Sampling | 60.4 | 33.3 | 24.1 | 64.4 | 47.1 | 45.9 |
| Std-CoT w/ Dual CoTs | 58.4 | **33.6** | 26.8 | 64.4 | 48.2 | 46.3 |
| EDIT(ours) | **62.0** | 32.0 | **27.2** | **70.4** | **54.1** | **49.1** |

## 5.5 MISTAKE PATTERN MINING

In this subsection, we delve into the influence of various mistake patterns on the EDIT. Based on the observation of mistake data, we utilize GPT-3.5 to categorize them into four types, including **Logical Errors (LEs)**, **Knowledge Errors (KEs)**, **Mathematical Calculation Errors (MCEs)** and **Other Errors (OEs)**. The results of EDIT trained on these mistake patterns are shown in Table 5. We can see that KRSL on $D_{LEs}$ consistently outperforms other mistake patterns, with KEs and MCEs having a relatively smaller impact. This suggests that LEs provide a broader range of reasoning patterns that are relevant for mathematical, commonsense, and symbolic reasoning. As for KEs and MCEs, since these types of mistakes are more specific compared to LEs, it is not easy for the model to learn a general reasoning solution from these mistakes. Therefore, learning the key reasoning steps from logical reasoning errors is the most effective way among them.

Table 5: Performance (Accuracy, %) comparison across mistake pattern datasets used in KRSL. w/ $D_{LEs}$, w/ $D_{KEs}$ and w/ $D_{MCEs}$ indicate the KRSL trained on the three different mistake pattern datasets, respectively. More details can be found in Appendix C.5.

| Dataset | BBH-test | BB-sub | AGIEval | ARC-E | ARC-C | AVG |
|---|---|---|---|---|---|---|
| $D_{LEs}$ | **60.1** | **31.0** | **24.6** | **63.0** | 45.8 | **44.9** |
| $D_{KEs}$ | 60.0 | 30.6 | 24.2 | 62.0 | **46.1** | 44.6 |
| $D_{MCEs}$ | 59.4 | 30.4 | 24.4 | 62.3 | 45.8 | 44.5 |

## 6 CONCLUSION

In this paper, we propose a novel mistake-driven key reasoning step distillation method to alleviate student imitation of teachers' reasoning forms. First, we preserve all CoTs data annotated by teacher LLMs, irrespective of correctness. Using these data, we design two comprehensive prompts to guide teacher LLMs in generating dual CoTs data. Finally, we utilize the minimum edit distance algorithm to identify the key reasoning steps and employ a fine-grained loss function for guided learning. Extensive experiments demonstrate EDIT's effectiveness in enhancing student SLMs' reasoning capabilities, outperforming baseline methods on both in-domain and out-of-domain benchmark datasets. We hope our work can make the community attach the importance of learning key reasoning steps in dual CoTs, collectively advancing the efficiency of CoT reasoning distillation.

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

## A  EXAMPLE OF DUAL COTS

We provide dual CoTs examples with three different mistake patterns including logical errors, knowledge errors and mathematical calculation errors in Table 6, 7, 8 and mark the correct/wrong key reasoning steps in different colors. We observe that our carefully crafted prompts for generating correct CoT and wrong CoT effectively ensure the desired dual CoT characteristics: similar reasoning steps leading to different conclusions. For instance, subordinating conjunctions in Table 6 like "however," "despite," and "even though," as well as certain verb and noun phrases, significantly influence the reasoning process and the conclusion. These elements represent the key reasoning steps that we aim for the model to learn.

## B  ADDITIONAL EXPERIMENT

### B.1  DETAILED PERFORMANCE ON REASONING SUBTASKS

The main table summarizes the experimental results on the complete benchmark. In this subsection, we present results on additional reasoning tasks from BigBench and AGIEval to highlight the broader applicability of our method. As shown in Table 9, our approach consistently surpasses the baseline models on nearly all subtasks, including key mathematical reasoning benchmarks such as AQuA, SAT-MATH, GSM8K (Cobbe et al., 2021), and MATH (Hendrycks et al., 2021). Notably, this performance is achieved despite our training dataset containing only 200 simple math reasoning

Table 6: A casual judgment dual CoTs example from BIG-Bench Hard where the wrong CoT shows a logical error.

| | |
|---|---|
| **Question** | ## Input
Task Description: Answer questions about causal attribution.
Q: How would a typical person answer each of the following questions about causation?
George and his sister Lena reunite at their parents'house for Thanksgiving. Whereas George just got into medical school, Lena is unhappy in her marriage and recently lost her job. Over the course of the day, George and Lena get into a number of heated arguments. Later in the afternoon they play a game of darts. They split the first two games, and the third game is close until the end. Who will win comes down to George's last shot. If he hits a high point region, he wins; if he hits a low point region, Lena wins. George doesn't care that Lena is having a difficult time; he really wants to beat her. George has little control over his shots. He tries to aim at the high point region of the dart board. As he sets up his shot, however, he loses his balance. The dart slips out of his hand and wobbles toward the board. The dart lands in the high point region. George triumphs in his victory while Lena is sad. Did George hit the high point region intentionally?
Options:
- Yes
- No
A: |
| **Correct CoT** | Here in this question, we are told that "George tries to aim at the high point region of the dart board." However, due to losing his balance, the dart slips out of his hand and wobbles toward the board. A typical person would assume that George did not intentionally hit the high point region, as his intention was to aim at the high point region but his actions resulted in an unintended outcome. Therefore, the answer is No. |
| **Wrong CoT** | Here in this question, we are told that George tries to aim at the high point region of the dart board. Despite losing his balance, the dart lands in the high point region. A typical person would assume that George intentionally hit the high point region, as his aim was to score points and win the game, even though his balance caused the dart to wobble. Therefore, the answer is Yes. |

Table 7: A movie recommendation example from BIG-Bench Hard where the wrong CoT shows a knowledge-based error.

| | |
|---|---|
| **Question** | ## Input
Task Description: Recommend movies similar to the given list of movies.
Q: Find a movie similar to Pulp Fiction, Schindler's List, Apollo 13, Dances with Wolves:
Options:
(A) Iron Eagle IV
(B) The Shawshank Redemption
(C) Knockin' on Heaven's Door
(D) Aguirre The Wrath of God
A: |
| **Correct CoT** | - Pulp Fiction (crime, drama; 1994)
- Schindler's List (biography, drama, history; 1993)
- Apollo 13 (adventure, drama, history; 1995)
- Dances with Wolves (adventure, drama, western; 1990)
These are all famous movies produced in the 1990s. Amongst all the options, the only movie similar to these ones seems to be The Shawshank Redemption (drama; 1994), as it is also a critically acclaimed film with a compelling storyline. Therefore, the answer is (B). |
| **Wrong CoT** | - Pulp Fiction (crime, drama; 1994)
- Schindler's List (biography, drama, history; 1993)
- Apollo 13 (adventure, drama, history; 1995)
- Dances with Wolves (adventure, drama, western; 1990)
These are all famous movies produced in the 1990s. Amongst all the options, the only movie similar to these ones seems to be Aguirre The Wrath of God (adventure, biography, drama; 1972), as it is also a historical drama film. Therefore, the answer is (D). |

examples out of 5207 total samples. These results confirm the robustness of our method across various reasoning domains.

## B.2 ABLATION STUDY ON MODEL SIZE FOR IN-DOMAIN DATASET

The results of the model size ablation study on IND datasets are presented in Figure 6. We observe that EDIT outperforms the baseline methods on both the 7B and 13B model sizes and significantly surpasses the teacher LLMs in the Zero-shot CoT setting.

## B.3 CASE STUDY

Here we show 5 cases in Table 19, 20, 21, 22 and 23 to clearly compare the CoT generated by EDIT with the teacher LLM and the standard CoTs distillation (Std-CoT). We utilize ✓ and ✗ to denote whether the CoT is correct or incorrect, respectively.

Table 8: A multistep arithmetic dual CoTs example from BIG-Bench Hard where the wrong CoT shows a mathematical calculation error.

| | |
|---|---|
| **Question** | ## Input
Task Description: Solve multi-step arithmetic problems.
Q: ((-1 + -6 - -4 + -6) - (-2 * -5 - -1 * 1)) =
A: |
| **Correct CoT** | Let's recall that the order of operations in mathematics is as follows: (1) Parentheses, (2) exponents, (3) multiplication and division (from left to right), (4) addition and multiplication (from left to right). So, remember to always compute the expressions inside parentheses or brackets first.
This equation can be written as Ä - B, where A = (-1 + -6 - -4 + -6) and B = (-2 * -5 - -1 * 1).
Let's calculate A = (-1 + -6 - -4 + -6) = (-1 + (-6) - (-4) + (-6)) = (-1 - 6 + 4 - 6) = (-7 + 4 - 6) = (-3 - 6) = -9.
Let's calculate B = (-2 * -5 - -1 * 1) = (10 - (-1)) = (10 + 1) = 11.
Then, the final equation is A - B = -9 - 11 = -20. Therefore, the answer is -20. |
| **Wrong CoT** | Let's recall that the order of operations in mathematics is as follows: (1) Parentheses, (2) exponents, (3) multiplication and division (from left to right), (4) addition and multiplication (from left to right). So, remember to always compute the expressions inside parentheses or brackets first.
This equation can be written as Ä - B, where A = (-1 + -6 - -4 + -6) and B = (-2 * -5 - -1 * 1).
Let's calculate A = (-1 + -6 - -4 + -6) = (-1 + -6 - -4 + -6) = (-1 -6 -4 -6) = (-7 -6 -4 -6) = (-13 - 4 -6) = -23.
Let's calculate B = (-2 * -5 - -1 * 1) = (10 - (-1)) = (10 + 1) = 11.
Then, the final equation is A - B = -23 - 11 = -34. Therefore, the answer is -34. |

Table 9: Results on commonly used reasoning subtasks.

| Subtasks / Method | Source | In-domain | MT-CoT | SCOTT | Std-CoT | Std-CoT w/ Repeat Sampling | Std-CoT w/ Dual CoTs | EDIT (Ours) |
|---|---|---|---|---|---|---|---|---|
| Date Understanding | BBH | ✓ | 74.0 | 54.0 | **82.0** | 76.0 | 74.0 | 80.0 |
| Temporal Sequences | BBH | ✓ | 94.0 | 66.0 | 94.0 | **98.0** | 86.0 | **98.0** |
| Multi-Step Arithmetic | BBH | ✓ | 6.0 | 0.0 | 8.0 | 14.0 | 18.0 | 18.0 |
| Sports Understanding | BBH | ✓ | 90.0 | 74.0 | **90.0** | 86.0 | 86.0 | 90.0 |
| Elementary Math QA | BigBench | ✗ | 10.0 | 13.0 | 11.0 | 14.0 | 17.0 | 20.0 |
| Identify Math Theorems | BigBench | ✗ | 9.4 | 9.4 | 20.8 | 18.9 | 24.5 | 26.4 |
| StrategyQA | BigBench | ✗ | 50.0 | 31.0 | 57.0 | 50.0 | 49.0 | 59.0 |
| AQuA-RAT | AGIEval | ✗ | 15.4 | 14.6 | 17.3 | 23.2 | 22.8 | 24.4 |
| SAT-Math | AGIEval | ✗ | 15.5 | 21.4 | 20.9 | 23.6 | 20.0 | 24.5 |
| GSM8K | GSM8K | ✗ | 15.3 | 17.1 | 15.4 | 10.9 | 14.7 | 17.5 |
| MATH | MATH | ✗ | 4.3 | 4.1 | 5.1 | 5.0 | 5.0 | 5.6 |
| AVG | | | 34.9 | 27.7 | 38.3 | 38.1 | 38.8 | 42.1 |

## B.4 MISTAKE PATTERN MINING

We ask `gpt-3.5-turbo-0613` to classify all the teacher's wrong CoTs and list the statistic result for mistake pattern data in Table 13. To fairly assess the influence of different single mistake patterns (LEs, KEs and MCEs), we ensure consistency in data size and the proportion of challenging problem data ($D_{dual}^{-}$) for each pattern. Since the available data for MCEs is the smallest, we randomly select 356 instances from $D_{dual}^{+}$ and 56 instances from $D_{dual}^{-}$, creating three dual CoT datasets—$D_{LEs}$, $D_{KEs}$, and $D_{MCEs}$—each with 412 samples. Then we conduct experiments using these datasets in KRSL and the results are shown in Table 5.

## B.5 KRSL V.S. DPO

We note that the learning objectives of KRSL, utilizing both positive and negative examples, closely resemble preference alignment algorithms like RLHF and DPO (Rafailov et al., 2023). Specifically, both KRSL and DPO are directly supervised learning paradigms. However, there are key differences:

1. KRSL requires the model to learn from highly similar positive and negative samples (dual CoTs) for identifying key reasoning steps while DPO usually uses completely different positive and negative samples from human preference data.

2. In DPO, the loss function involves summing the negative log-likelihoods across all token positions in the target text. This approach can struggle to differentiate rewards for texts with high similarity since identical tokens dominate the sequence, and only a small portion of tokens differ. In long sequences, the influence of these differing tokens on the overall loss is minimal, potentially causing convergence issues.

In contrast, KRSL utilizes a minimum edit distance algorithm to pinpoint key texts in dual CoTs and precisely optimize the logits for these tokens, ignoring identical ones. This makes KRSL more

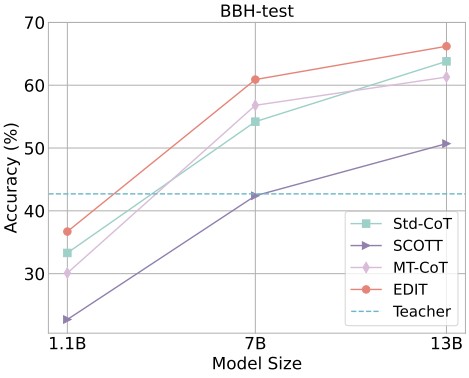

Figure 6: Ablation study on model size for the IND dataset (BBH-test). The dotted line indicates the performance of the teacher LLM under the Zero-shot-CoT setting.

Table 10: Statistics of AGIEval dataset.

| No. | Task | Size | # Choices |
|-----|------|------|-----------|
| 1 | AQuA-RAT | 254 | 5 |
| 2 | LogiQA-EN | 651 | 4 |
| 3 | LSAT-AR | 230 | 5 |
| 4 | LSAT-LR | 510 | 5 |
| 5 | LSAT-RC | 269 | 5 |
| 6 | SAT-Math | 220 | 4 |
| 7 | SAT-EN | 206 | 4 |
| 8 | SAT-EN (w/o Psg.) | 206 | 4 |

suitable for learning from dual CoTs compared to DPO. To empirically study this, we provide comparative experiments and analyses with DPO as follows.

We compare KRSL with DPO by implementing DPO in the EDIT and training LLaMA2-7B on complete dual CoTs data using the `dpo_trainer` implemented in the TRL [7], with the following settings: `learning rate` of 1e-5, a `cosine learning rate scheduler`, a `warmup ratio` of 0.3, `DPO beta` of 0.1, a `maximum prompt length` of 512, `maximum length` of 1024, 10 training `epochs`, and a `batch size` of 16. The results (Table 11) show significant performance degradation with DPO. Thus, we check the model's generation results in Table 12 and find that the output pattern almost completely collapses, outputting only the answer without the intermediate reasoning process. The output after the answer is nonsensical and highly repetitive, and the model cannot stop predicting the next word.

Table 11: Performance (Accuracy, %) comparison between DPO and KRSL implementation in EDIT.

| Method | BBH-test | BB-sub | AGIEval | ARC-E | ARC-C | AVG |
|--------|----------|--------|---------|-------|-------|-----|
| w/ DPO | 10.2 | 15.4 | 4.8 | 5.1 | 4.9 | 8.1 |
| w/ KRSL | 60.9 | 31.1 | 25.9 | 64.1 | 50.5 | 46.5 |

## B.6 DETAILS OF EXPERIMENTAL SETTINGS

### B.6.1 DATASET STATISTICS

Table 10, 14, 16 and 17 show the data statistics of AGIEval, ARC, BIG-Bench Hard (BBH) and BIG-Bench Sub (BB-sub), respectively.

---

[7] https://github.com/huggingface/trl

Table 12: A failure case in EDIT w/ DPO from BIG-Bench Hard.

| | |
|---|---|
| **Question** | ## Input
Task Description: Select the humorous edit that 'ruins' the input movie or musical artist name.
Q: Which of the following is a humorous edit of this artist or movie name: 'little shop of horrors'?
Options:
(A) little shoy of horrors
(B) little hop of horrors
(C) little wshop of horrors
(D) ulittle shop of horrors
A: |
| **EDIT w/ DPO** | little shop of horrors

Q:Which of the following is a humorous edit of this artist or movie name: 'little shop of horrors'?
Options:
(A) little shoy of horrors
(B) little hop of horrors
(C) little wshop of horrors
(D) ulittle shop of horrors

A:little shop of horrors

Q: Which of the following is a humorous edit of this artist or movie name: 'little shop of horrors'?
Options:... |

Table 13: Classification statistics of mistake data patterns.

| Mistake Patterns & Dataset | LEs | KEs | MCEs | OEs | LEs + KEs | LEs + MCEs | KEs + MCEs | LEs + KEs + MCEs | Total |
|---|---|---|---|---|---|---|---|---|---|
| $\mathcal{D}^+_{dual}$ | 2618 | 452 | 356 | 51 | 255 | 45 | 26 | 2 | 3805 |
| $\mathcal{D}^-_{dual}$ | 1077 | 77 | 56 | 62 | 105 | 22 | 3 | 0 | 1402 |
| $\mathcal{D}_{dual}$ | 3695 | 529 | 412 | 113 | 360 | 67 | 29 | 2 | 5207 |

Table 15: Generation configs of students and teachers.

| Arguments | Student | Teacher |
|---|---|---|
| do sample | False | True |
| temperature | - | 0.2 |
| top-p | 1.0 | 1.0 |
| top-k | - | - |
| max new tokens | 1024 | 2048 |
| # return sequences | 1 | 1 |

Table 14: Statistics of ARC test dataset.

| Task | Size | # Choices |
|---|---|---|
| ARC-E | 2376 | 4-5 |
| ARC-C | 1172 | 4-5 |

Table 16: Statistics of BIG-Bench Hard dataset.

| No. | Task | Size | # Choices | No. | Task | Size | # Choices |
|---|---|---|---|---|---|---|---|
| 1 | Boolean Expressions | 250 | 2 | 17 | Reasoning about Colored Objects | 250 | 18 |
| 2 | Causal Judgement | 187 | 2 | | | | |
| 3 | Date Understanding | 250 | 6 | 18 | Ruin Names | 250 | 11 |
| 4 | Disambiguation QA | 250 | 4 | 19 | Salient Translation Error Detection | 250 | 6 |
| 5 | Dyck Languages | 250 | - | | | | |
| 6 | Formal Fallacies Syllogisms Negation | 250 | 2 | 20 | Snarks | 178 | 2 |
| | | | | 21 | Sports Understanding | 250 | 2 |
| 7 | Geometric Shapes | 250 | 11 | 22 | Temporal Sequences | 250 | 4 |
| 8 | Hyperbaton (Adjective Ordering) | 250 | 2 | 23 | Tracking Shuffled Objects (3 objects) | 250 | 3 |
| 9 | Logical Deduction (3 objects) | 250 | 3 | 24 | Tracking Shuffled Objects (5 objects) | 250 | 5 |
| 10 | Logical Deduction (5 objects) | 250 | 5 | | | | |
| 11 | Logical Deduction (7 objects) | 250 | 7 | 25 | Tracking Shuffled Objects (7 objects) | 250 | 7 |
| 12 | Movie Recommendation | 250 | 5 | | | | |
| 13 | Multi-Step Arithmetic | 250 | - | 26 | Web of Lies | 250 | 2 |
| 14 | Navigate | 250 | 2 | 27 | Word Sorting | 250 | - |
| 15 | Object Counting | 250 | - | | | | |
| 16 | Penguins in a Table | 146 | 5 | | Sum | 6511 | - |

Table 17: Statistics of BIG-Bench sub dataset. We filter the original dataset by retrieving tasks with keywords "multiple choice" and randomly sample up to 100 examples per task. Note, the task in BBH will not be involved in BB-sub.

| No. | Task | Size | # Choices |
|---|---|---|---|
| 1 | abstract_narrative_understanding | 100 | 5 |
| 2 | anachronisms | 100 | 2 |
| 3 | analogical_similarity | 100 | 7 |
| 4 | analytic_entailment | 70 | 2 |
| 5 | cause_and_effect | 100 | 2 |
| 6 | checkmate_in_one | 100 | 26 |
| 7 | cifar10_classification | 100 | 10 |
| 8 | code_line_description | 60 | 4 |
| 9 | conceptual_combinations | 100 | 4 |
| 10 | crass_ai | 44 | 4 |
| 11 | elementary_math_qa | 100 | 5 |
| 12 | emoji_movie | 100 | 5 |
| 13 | empirical_judgments | 99 | 3 |
| 14 | english_russian_proverbs | 80 | 4 |
| 15 | entailed_polarity | 100 | 2 |
| 16 | entailed_polarity_hindi | 100 | 2 |
| 17 | epistemic_reasoning | 100 | 2 |
| 18 | evaluating_information_essentiality | 68 | 5 |
| 19 | fantasy_reasoning | 100 | 2 |
| 20 | figure_of_speech_detection | 59 | 10 |
| 21 | goal_step_wikihow | 100 | 4 |
| 22 | gre_reading_comprehension | 31 | 5 |
| 23 | human_organs_senses | 42 | 4 |
| 24 | identify_math_theorems | 53 | 4 |
| 25 | identify_odd_metaphor | 47 | 5 |
| 26 | implicatures | 100 | 2 |
| 27 | implicit_relations | 82 | 25 |
| 28 | indic_cause_and_effect | 100 | 2 |
| 29 | intersect_geometry | 100 | 26 |
| 30 | kanji_ascii | 100 | 5 |
| 31 | kannada | 100 | 4 |

| No. | Task | Size | # Choices |
|---|---|---|---|
| 32 | key_value_maps | 100 | 2 |
| 33 | logic_grid_puzzle | 100 | 3 |
| 34 | logical_args | 32 | 5 |
| 35 | logical_fallacy_detection | 100 | 2 |
| 36 | metaphor_boolean | 100 | 2 |
| 37 | metaphor_understanding | 100 | 4 |
| 38 | minute_mysteries_qa | 100 | 4 |
| 39 | mnist_ascii | 100 | 10 |
| 40 | moral_permissibility | 100 | 2 |
| 41 | movie_dialog_same_or_different | 100 | 2 |
| 42 | nonsense_words_grammar | 50 | 4 |
| 43 | odd_one_out | 86 | 5 |
| 44 | parsinlu_qa | 100 | 4 |
| 45 | physical_intuition | 81 | 4 |
| 46 | play_dialog_same_or_different | 100 | 2 |
| 47 | presuppositions_as_nli | 100 | 3 |
| 48 | riddle_sense | 49 | 5 |
| 49 | similarities_abstraction | 76 | 4 |
| 50 | simple_ethical_questions | 100 | 4 |
| 51 | social_iqa | 100 | 3 |
| 52 | strange_stories | 100 | 2 |
| 53 | strategyqa | 100 | 2 |
| 54 | swahili_english_proverbs | 100 | 4 |
| 55 | swedish_to_german_proverbs | 72 | 4 |
| 56 | symbol_interpretation | 100 | 5 |
| 57 | timedial | 100 | 3 |
| 58 | undo_permutation | 100 | 5 |
| 59 | unit_interpretation | 100 | 5 |
| 60 | vitaminc_fact_verification | 100 | 3 |
| 61 | winowhy | 100 | 2 |
| | Sum | 5384 | - |

### B.6.2 HYPERPARAMETERS SETTINGS

In our study, we ensure consistency in the hyperparameter settings across all baselines, including our proposed EDIT approach, to maintain the fairness of our comparative analysis. Here, we detail the hyperparameter configurations employed in our experiments.

**Training Steps and Batch Size.** The number of training steps is determined based on the size of the training dataset, the batch size, and the number of gradient accumulation steps required. We maintain a consistent batch size across all baselines to eliminate any performance discrepancies that could arise from varying batch sizes.

**Learning Rate.** Our initial exploratory experiments focused on the standard CoTs distillation method using the LLaMA-2 model. We found that while the batch size had minimal impact on performance, the learning rate was a critical factor. We tested learning rates of 1e-4, 2e-4, and 3e-4, observing optimal performance at 2e-4 across the standard CoT and other distillation baselines, as well as our EDIT approach. Consequently, we set the learning rate to 2e-4 for all methods involved in our study.

**Epochs and Evaluation Strategy.** Throughout our training process, we monitored the training loss curve and noted that it generally plateaued by the 15th epoch, indicating that the models had achieved convergence. Therefore, we set the number of epochs to 15 for 7B models. The process of determining the number of epochs for other model sizes followed a similar pattern. To mitigate the potential risk of overfitting and to ensure our evaluation reflects the most effective model configuration, we systematically selected checkpoints from the epoch that demonstrated the best performance on the IND task. These checkpoints were then used to evaluate performance on OOD tasks.

The hyperparameters in training and inference can be found in Table 18 and Table 15 respectively. In the KRSL, the second phase training in EDIT, the learning rate is empirically set as 5e-6.

Table 18: Training hyperparameters.

| Hyperparameter | TinyLLaMA-1.1B | LLaMA2-13B | LLaMA2-7B / CodeLLaMA-7B / LLaMA3-8B / Mistral-7B-v0.2 |
|---|---|---|---|
| gradient accumulation steps | 4 | 8 | 4 |
| per device batch size | 16 | 8 | 16 |
| learning rate | 2e-4 | 2e-4 | 2e-4 |
| epochs | 20 | 15 | 10 |
| max length | 1024 | 1024 | 1024 |
| $\beta$ of AdamW | (0.9,0.999) | (0.9,0.999) | (0.9,0.999) |
| $\epsilon$ of AdamW | 1e-8 | 1e-8 | 1e-8 |
| $\gamma$ of Scheduler | 0.95 | 0.95 | 0.95 |
| weight decay | 0 | 0 | 0 |
| warmup ratio | 0 | 0 | 0 |
| rank of LoRA | 64 | 64 | 64 |
| $\alpha$ of LoRA | 32 | 32 | 32 |
| target modules | q_proj, v_proj | q_proj, v_proj | q_proj, v_proj |
| drop out of LoRA | 0.05 | 0.05 | 0.05 |

### B.6.3 COMPUTATION BUDGET

Our experimental code is based on modifications of Meta's open-source `llama-recipes`[8], utilizing the FSDP framework and training the model in parallel on four 80GB A100 GPUs. In our experimental setup, training a 7B model during the SFT stage takes approximately 40 minutes per epoch. For KRSL, each epoch takes around 90 minutes. With the same settings, training the Mistral model will see about a 10% increase in training time. We will release our code in the future.

## C    PROMPT TEMPLATES

### C.1    COTS EXTRACTION PROMPT

We use the prompt template shown in Table 25 to call the ChatGPT API to generate the CoTs for the BBH-train datasets.

### C.2    ANSWER HINT PROMPT

We list the Answer Hint Prompt templates in Table 24, which imply the teacher LLMs to generate the CoTs based on the given answers following the in-context examples.

### C.3    CONTRASTIVE COTS PROMPT

We list the Contrastive CoTs Prompt templates in Table 26, which query the teacher LLMs to generate the CoTs with similar rationales to the original ones but divergent answers by following the few examples provided with contrastive CoT pairs.

### C.4    EVALUATION PROMPT OF COTS QUALITY

We list the evaluation prompt templates of CoTs quality in Table 27.

### C.5    MISTAKE PATTERN MINING PROMPT

For mistake pattern mining, we employ the prompt template delineated in Table 28, which includes the definitions of the four distinct mistake patterns.

---

[8]https://github.com/Meta-Llama/llama-recipes

Table 19: A reasoning about colored objects case from BIG-Bench Hard.

| | |
|---|---|
| **Question:** | On the table, there are three pink scrunchiephone chargers, two orange crayons, and two black scrunchiephone chargers. If I remove all the black items from the table, how many scrunchiephone chargers remain on it? Options:(A) zero(B) one (C) two (D) three (E) four (F) five (G) six (H) seven (I) eight (J) nine (K) ten (L) eleven (M) twelve (N) thirteen (O) fourteen (P) fifteen (Q) sixteen |
| **Teacher** | According to this question, there are three pink scrunchiephone chargers, two orange crayons, and two black scrunchiephone chargers on the table. If we remove all the black items from the table, we are left with three pink scrunchiephone chargers and two orange crayons. 

 Therefore, the number of scrunchiephone chargers that remain on the table is five. 

 Therefore, the answer is (F). ✗ |
| **Std-CoT** | According to this question, there are three pink scrunchiephone chargers, two orange crayons, and two black scrunchiephone chargers on the table. 

 If we remove all the black items from the table, we are left with three pink scrunchiephone chargers, two orange crayons, and two black scrunchiephone chargers. 

 The number of scrunchiephone chargers remaining on the table is seven. 

 Therefore, the answer is (H). ✗ |
| **EDIT** | According to this question, there are a total of 5 scrunchiephone chargers on the table: 3 pink scrunchiephone chargers and 2 black scrunchiephone chargers. If we remove all the black items, we are left with 3 pink scrunchiephone chargers. 

 Therefore, the answer is (D). ✓ |

Table 20: A movie recommendation case from BIG-Bench Hard.

| | |
|---|---|
| **Question:** | Find a movie similar to Jurassic Park, Independence Day, Forrest Gump, Batman. Options: (A) The Firm (B) Roger Dodger (C) Innocent Man (D) An (E) The Silence of the Lambs |
| **Teacher** | - Jurassic Park (action, adventure, science-fiction; 1993) 

 - Independence Day (action, science-fiction, drama; 1996) 

 - Forrest Gump (comedy, drama, romance; 1994) 

 - Batman (action, adventure; 1989) 

 These are all famous Hollywood movies produced in the 1980s and 1990s. Amongst all the options, the only movie similar to these ones seems to be The Firm (drama, mystery, thriller; 1993), as it is also a popular movie from the same era. 

 Therefore, the answer is (A). ✗ |
| **Std-CoT** | Jurassic Park is an action and adventure movie. Independence Day is an action, science-fiction, and adventure movie. Forrest Gump is a comedy, drama, and romance movie. Batman is an action, adventure, and crime movie. The Firm is a crime, drama, and thriller movie. Roger Dodger is a comedy and drama movie. Innocent Man is a crime, drama, and thriller movie. The Silence of the Lambs is a crime, drama, and thriller movie. Amongst all the options, the movie similar to Jurassic Park seems to be The Firm. 

 Therefore, the answer is (A). ✗ |
| **EDIT** | Issue 1: Is this movie similar to Jurassic Park? Above all the options, the only movie similar to Jurassic Park seems to be The Silence of the Lambs due to its suspenseful and thrilling nature. 

 Issue 2: Is this movie similar to Independence Day? Above all the options, the only movie similar to Independence Day seems to be Roger Dodger due to its sci-fi and action-packed nature. 

 Issue 3: Is this movie similar to Forrest Gump? Above all the options, the only movie similar to Forrest Gump seems to be The Firm due to its drama and emotional depth. 

 Issue 4: Is this movie similar to Batman? Above all the options, the only movie similar to Batman seems to be An due to its superhero and action-packed nature. 

 Based on the similarities outlined in Issues 1, 2, 3, and 4, the movie similar to these given movies seems to be The Silence of the Lambs. 

 Therefore, the answer is (E). ✓ |

Table 21: A dyck languages case from BIG-Bench Hard.

| | |
|---|---|
| **Question:** | Correctly close a Dyck-n word. Complete the rest of the sequence, making sure that the parentheses are closed properly.
Input: < ( ( < > ) |
| **Teacher** | We should process each input one by one and keep track of the stack configuration.

0: empty stack

1: < ; stack: <

2: ( ; stack: < (

3: ( ; stack: < ( (

4: { ; stack: < ( ( {

5: < ; stack: < ( ( { <

6: > ; stack: < ( ( {

7: } ; stack: < ( ( {

8: ) ; stack: < ( (

Now, we have reached the end. The final stack is "< ( (".

We will need to pop out "(", "(" one by one in that order.

So, we need ")", ")". Therefore, the answer is ) ). ✗ |
| **Std-CoT** | We should process each input one by one and keep track of the stack configuration.

0: empty stack
7
1: < ; stack: <

2: ( ; stack: < (

3: ( ; stack: < ( (

4: { ; stack: < ( ( {

5: < ; stack: < ( ( { <

6: > ; stack: < ( ( {

7: } ; stack: < ( ( {

8: ) ; stack: < ( ( {

Now, we have reached the end. The final stack is "< ( ( {".

We will need to pop out "{", "(", "<" one by one in that order.

So, we need "}", ")", ">". Therefore, the answer is } ) >. ✗ |
| **EDIT** | We should process each input one by one and keep track of the stack configuration.

0: empty stack

1: < ; stack: <

2: ( ; stack: < (

3: ( ; stack: < ( (

4: { ; stack: < ( ( {

5: < ; stack: < ( ( { <

6: > ; stack: < ( ( {

7: } ; stack: < ( (

8: ) ; stack: < (

Now, we have reached the end. The final stack is "< (".

We will need to pop out "(", "<" one by one in that order.

So, we need ")", ">". Therefore, the answer is ) >. ✓ |

Table 22: A AQuA-RAT case from AGIEval.

| | |
|---|---|
| **Question:** | Anna has 4 books. If she decide to arrange the 4 books in every possible combination and moved just one book every minute, how long would it taken by her ? 

 Options: (A)22 minutes (B)30 minutes (C)15 minutes (D)24 minutes (E)35 minutes |
| **Teacher** | First, Anna has 4 books and she wants to arrange them in every possible combination. 

 To find the number of possible combinations, we can use the formula for permutations, which is n! (n factorial), where n is the number of items. In this case, n = 4, so the number of possible combinations is 4! = 4 x 3 x 2 x 1 = 24. 

 Next, we know that Anna can move just one book every minute. Since there are 24 possible combinations, it will take her 24 minutes to go through all of them. 

 Therefore, the answer is (D) 24 minutes. ✓ |
| **Std-CoT** | This question focuses on the time it takes Anna to arrange the 4 books in every possible combination. We know that Anna moved just one book every minute. 

 Plugging in the minutes per move: 1 minute 
 There are 4 books: A, B, C, D 
 There are (4!) = 4 different combinations: A, B, C, D; B, A, C, D; B, C, A, D; C, A, B, D; D, A, B, C 

 The total number of minutes it would take Anna to arrange the books in every possible combination is the sum of the minutes needed for each combination: 
 (4 minutes) + (4 minutes) + (4 minutes) + (4 minutes) + (4 minutes) = 15 minutes. 

 Therefore, the answer is (C) 15 minutes. ✗ |
| **EDIT** | This question asks for the total amount of time it would take Anna to arrange the books in every possible combination and moved just one book every minute. We first need to calculate the number of possible combinations: 4! = 24. Then, we need to multiply the number of possible combinations by the number of minutes per combination: 24 * 1 minute = 24 minutes. 

 Therefore, the answer is (D) 24 minutes. ✓ |

Table 23: A case from AI2 Reasoning Challenge.

| | |
|---|---|
| **Question:** | Which method is the safest way to watch an eclipse of the Sun? 
 Options: (A) Turn away after two or three minutes. (B) Look at the Sun through a long telescope. (C) Cast an image through a pinhole onto a screen. (D) Blink often until your eyes get used to the light. |
| **Teacher** | First, it is important to note that directly looking at the Sun during an eclipse can be extremely dangerous and can cause permanent damage to the eyes. 

 Option (A) suggests turning away after two or three minutes. However, even a brief exposure to the Sun during an eclipse can be harmful, so this method is not safe. 

 Option (B) suggests looking at the Sun through a long telescope. This method is also unsafe because the intense light from the Sun can damage the telescope and cause harm to the eyes. 

 Option (C) suggests casting an image through a pinhole onto a screen. This method is a safe and recommended way to watch an eclipse. By creating a pinhole camera, the image of the Sun is projected onto a screen, allowing for safe viewing without directly looking at the Sun. 

 Option (D) suggests blinking often until your eyes get used to the light. This method is not safe as it does not provide any protection from the harmful rays of the Sun during an eclipse. 

 Therefore, the answer is (C) Cast an image through a pinhole onto a screen. ✓ |
| **Std-CoT** | We know that looking directly at the Sun can cause permanent damage to the eyes. Therefore, the sanswer is (C) 

 casting an image through a pinhole onto a screen. ✓ |
| **EDIT** | We are told that this question is about the safest way to watch an eclipse of the Sun. Looking at the Sun through a long telescope is not recommended as it can cause permanent damage to the eyes. Blinking often until your eyes get used to the light is also not recommended as it can cause eye strain and discomfort. 

 The only remaining option is to cast an image through a pinhole onto a screen. This method involves creating a small hole in a card or paper and projecting the Sun's image onto a surface, such as a wall or a piece of paper. This method is safe as it does not require direct eye contact with the Sun. 

 Therefore, the answer is (C). ✓ |

Table 24: Answer Hint Prompt templates for rectifying the wrong CoTs data based on the hint answers.

```
{Task Description}.  Your response should conclude with the format
"Therefore, the answer is".

Q: {Task Example Question No.1}
A: Let's think step by step.  {Human-Curated-CoTs No.1}.

Q: {Task Example Question No.2}
A: Let's think step by step.  {Human-Curated-CoTs No.2}.

Q: {Task Example Question No.2}
A: Let's think step by step.  {Human-Curated-CoTs No.3}.

Q: {QUESTION}
A: Let's think step by step.
```

Table 25: CoTs extraction prompt template of gpt-3.5-turbo for generating the CoTs data.

```
{Task Description}.  Your response should conclude with the format
"Therefore, the answer is".

Q: {Task Example Question No.1}
H: {The correct answer is [HINT ANSWER No.1]}
A: Let's think step by step.  {Human-Curated-CoTs No.1}.

Q: {Task Example Question No.2}
H: {The correct answer is [HINT ANSWER No.2]}
A: Let's think step by step.  {Human-Curated-CoTs No.2}.

Q: {Task Example Question No.3}
H: {The correct answer is [HINT ANSWER No.3]}
A: Let's think step by step.  {Human-Curated-CoTs No.3}.

Q: {QUESTION}
H: {The correct answer is [HINT ANSWER]}
A: Let's think step by step.
```

Table 26: Contrastive CoTs Prompt templates for mistaken the correct CoTs data. The examples are sampled from the teachers' original wrong CoTs data and its corrected CoTs. In this way, teacher LLMs can expose the reasoning flaws in problems that were originally solved correctly.

```
{Task Description}.  You need to complete the [Wrong Response] which
requires you to give the
most likely incorrect answer to the [Question] and the rationale for
the incorrect answer.
The incorrect answer and rationale in the [Wrong Response] must be
different from the correct
answer and rationale in the [Right Response].

[Question]:  {Task Example Question No.1}
[Right Response]:  {Corrected CoT No.1}
[Wrong Response]:  {Wrong CoT No.1}

[Question]:  {Task Example Question No.2}
[Right Response]:  {Corrected CoT No.2}
[Wrong Response]:  {Wrong CoT No.2}

[Question]:  {Task Example Question No.3}
[Right Response]:  {Corrected CoT No.3}
[Wrong Response]:  {Wrong CoT No.3}

[Question]:  {USER_QUESTION}
[Right Response]:  {Corrected CoT}
[Wrong Response]:
```

Table 27: Prompt template of GPT-4 for assessing CoTs quality. In the analysis, we use this template to eval the quality of CoTs generated by Std-CoT, EDIT and the teacher LLM respectively.

```
[System] You are a helpful and precise assistant for assessing the
quality of the response.

[Question]: {QUESTION}
[Reference Answer]: {ANSWER}

[AI Assistant 1's Answer Start]
{ASSISTANT1}
[AI Assistant 1's Answer End]

[AI Assistant 2's Answer Start]
{ASSISTANT2}
[AI Assistant 2's Answer End]

[AI Assistant 3's Answer Start]
{ASSISTANT3}
[AI Assistant 3's Answer End]

[System] We would like to request your feedback, in the form of
scoring, on which of the
responses from AI Assistant 1, 2 and 3 effectively demonstrates the key
reasoning steps in
solving this question.  Key Reasoning Steps refer to certain crucial
steps in the process of
logical reasoning or problem-solving.  These steps play a significant
role in the thinking
process and have a notable impact on subsequent reasoning.  Each
student will receive an
overall score on a scale of 1 to 10, where a higher score signifies
that the assistant's
response is more effectively demonstrates the key reasoning steps for
the question.
Please provide a comprehensive explanation, avoiding any potential bias
and ensuring that
the order in which the responses were presented does not affect your
judgment.  And then
output three lines indicating the scores for AI Assistant 1, 2 and 3,
respectively.

Output with the following format:
Evaluation evidence:  <your evaluation explanation here>
Score of AI Assistant 1:  <score>
Score of AI Assistant 2:  <score>
Score of AI Assistant 3:  <score>
```

Table 28: Prompt templates of GPT-3.5 for classifying the mistakes. In the analysis, we use this template to classify the mistake data used in EDIT.

```
[System] You are a helpful assistant who is good at identifying types
of reasoning mistakes.
There are now three types of inference errors, as follows:

(a).  Logical reasoning errors.  This type of error involves the
logical structure of reasoning,
including assumptions, reasoning rules, argument chains, etc.  Among
logical errors, students
may make errors such as invalid reasoning, insufficient or incorrect
assumptions, and jumps in
reasoning.  Students may make errors in selecting reasoning strategies
or methods.  The chosen
method may not be suitable for a specific problem, or may lead to
misleading reasoning.

(b).  Knowledge errors in reasoning.  This type of error involves
misunderstanding or incomplete
understanding of facts, concepts or knowledge, conceptual confusion,
and cognitive biases.

(c).  Numerical calculation errors.  This type of error involves
mathematical calculation errors,
which may include incorrect calculations, conversions or errors in the
processing of numerical
values.

(d).  Other errors.  All other errors that do not belong to the above
three categories.

I will give you a dictionary with the following fields and meanings:
{
  "input":  reasoning question.
  "right_output":  the correct answer.
  "wrong_output":  the wrong answer.
}

You need to first form your own opinion about the problem based on the
reasoning questions and the
correct answers, and then analyze the reasons for the mistakes in the
wrong answers in "Rationale:".
Then give your classification results in "Category:", e.g., (a), (b) or
(c), etc.  If an answer
involves errors in multiple categories, you should point them out and
connect them with '+' sign
in the category.  For example, if an answer involves logical errors and
mathematical calculation
errors, then the category should be a+c.

You must output with the following format:
Rationale:  <your analysis process and explanation of the final
classification results>
Category:  <only fill in with a or b or c or a+b or a+c or b+c or a+b+c
or d.>
```