# OpenReview forum: "Beyond Imitation: Learning Key Reasoning Steps from Dual Chain-of-Thoughts in Reasoning Distillation"
_ICLR.cc/2025/Conference — ICLR 2025 Conference Withdrawn Submission_

### Official Review · Reviewer_7kvH · 2024-10-31

**Soundness:** 2
**Presentation:** 3
**Contribution:** 2
**Rating:** 3
**Confidence:** 4

**Summary:**

This paper propose “Mistake-driven key reasoning step distillation”(EDIT) which aims to improve the reasoning capability of SLMs, focusing specifically on key reasoning steps. Unlike previous standard SFT techniques, EDIT fine-tunes SLMs solely on tokens within key reasoning steps, disregarding training loss for tokens that are not in key reasoning steps. To obtain the data for SFT, this paper prompts powerful LLMs to generate dual CoTs data by correcting erroneous CoTs and introducing errors into correct ones. Empirical results demonstrate the effectiveness of EDIT in significantly improving SLM performance on reasoning tasks.

**Strengths:**

1. EDIT innovatively fine-tunes SLMs solely on tokens that are different in dual CoT reasoning steps to prevent simply mimicking the reasoning form, which is resource-efficient and reasonable.
2. The paper is well-written and easy to follow.

**Weaknesses:**

1. Teacher LLMs generated contents lack evaluation, e.g, the correctness of reasoning steps after applying AHP to rectify wrong CoTs, and the ratio of correctly corrupted key reasoning steps?
2. The experiments of this paper are relatively weak. First, the compared baselines are all works essentially accomplished in 2022. Are there any brand new methods in 2023-2024? Second, the paper conducts in-domain experiments on only one dataset (BBH) which is released in 2022. It’s better to provide at least one more in-domain experiment on some convicing benchmarks to demonstrate the effectiveness of EDIT.
3. The structure of the paper can be optimized. I think it’s better to move some the results from Figure.5(middle) and Table.9 in Appendix to Table.1, making your main results more convicing. Figure 5 (right) is referenced in Section 5, yet it is included in the same figure as the content from Section 4. I believe it would be better to format it separately under Section 5.

**Questions:**

See Weaknesses.1.

---

> ### Author Response · Authors · 2024-11-22
>
> Thank you for your thorough and constructive comments. We sincerely appreciate the time and effort you have put into reviewing our work. Below, we address your concerns point by point:
>
> > **W1:** Teacher LLMs generated contents lack evaluation, e.g, the correctness of reasoning steps after applying AHP to rectify wrong CoTs, and the ratio of correctly corrupted key reasoning steps?
>
> **A1:** In our manual random check of 100 examples, approximately 93 dual CoTs exhibited reasoning processes that supported the final conclusion, indicating the correctness of their reasoning steps. The strong contextual learning capability and autoregressive nature of the Teacher LLM enable it to generate CoTs with reasoning processes that are similar yet lead to different conclusions and remain logically consistent. Additionally, in the final dual CoTs dataset, the proportion of key reasoning steps within the entire CoT sequence was statistically found to average approximately 4.7% (Line 53).
>
> > **W2:** The experiments of this paper are relatively weak. First, the compared baselines are all works essentially accomplished in 2022. Are there any brand new methods in 2023-2024? Second, the paper conducts in-domain experiments on only one dataset (BBH) which is released in 2022. It’s better to provide at least one more in-domain experiment on some convicing benchmarks to demonstrate the effectiveness of EDIT.
>
> **A2:**
>
> * We additionally compared our method against recent approaches [1, 2]. Hsieh et al. (2023) [1] proposed a multi-task CoTs distillation method that distills rationales and answers separately. Chen et al. (2024) [2] proposed to learn the mutual relationship of the two tasks from an Information Bottleneck perspective.From the experimental results in the table below, we observe that the EDIT method outperforms these baselines on three benchmarks, particularly for in-domain tasks. Although these methods excel in out-of-domain tasks, it is important to note that their evaluation only requires generating the answer. Since the learning objectives of these methods include directly fitting to the answer labels, this approach often results in low logical consistency between the rationale and answer generated by the student model. In contrast, our method is an improvement based on Std-CoT, where the reasoning generation mode ensures that the rationale and answer are coherent, thereby guaranteeing the fidelity of reasoning.
>
>   | Method | BBH-test | BB-sub | AGIEval | ARC-E | ARC-C | AVG |
>   | --- | --- | --- | --- | --- | --- | --- |
>   | In-domain | √   | ×   | ×   | ×   | ×   |     |
>   | MT-CoT | 56.8 | 30.3 | 22.0 | 49.4 | 38.2 | 39.3 |
>   | SCOTT | 42.4 | 18.8 | 13.0 | 45.7 | 34.1 | 30.8 |
>   | Std-CoT | 54.2 | 28.7 | 21.6 | 59.6 | 45.1 | 41.8 |
>   | Hsieh et al. (2023)[1] | 42.4 | 27.7 | 28.8 | **68.5** | 48.6 | 43.2 |
>   | Chen et al. (2024)[2] | 42.9 | 24.3 | **29.2** | 68.4 | 49.3 | 42.8 |
>   | EDIT(ours) | **60.9** | **31.1** | 25.9 | 64.1 | **50.5** | **46.5** |
>
> * Regarding in-domain experiments:Our original goal was to enhance the general reasoning ability of the student model. Therefore, we selected BBH, a challenging dataset encompassing 27 diverse reasoning tasks, as the in-domain dataset. Using the EDIT method, the model trained on the BBH data achieved significantly better performance on the test set of these 27 tasks compared to the baselines (+6.7%). We believe this sufficiently demonstrates the effectiveness of EDIT.
>
>
> [1] Hsieh C Y, Li C L, Yeh C, et al. Distilling Step-by-Step! Outperforming Larger Language Models with Less Training Data and Smaller Model Sizes[C]//Findings of the Association for Computational Linguistics: ACL 2023. 2023: 8003-8017.
>
> [2] Chen X, Huang H, Gao Y, et al. Learning to Maximize Mutual Information for Chain-of-Thought Distillation[C]//Findings of the Association for Computational Linguistics: ACL 2024. 2024: 6857-6868.
>
> > **W3:** The structure of the paper can be optimized. I think it’s better to move some the results from Figure.5(middle) and Table.9 in Appendix to Table.1, making your main results more convicing. Figure 5 (right) is referenced in Section 5, yet it is included in the same figure as the content from Section 4. I believe it would be better to format it separately under Section 5.
>
> **A3:** Thank you for carefully reading our paper and providing guidance on its structure. We will adopt your suggestions to optimize the organization of the paper. Specifically:
>
> * We will move relevant results from Figure 5 (middle) and Table 9 in the Appendix to Table 1 to make the main results more prominent and convincing.
> * We will reformat Figure 5 (right) to place it under Section 5 to ensure a clear and logical presentation.
>
> We greatly value your thoughtful feedback and are committed to improving our work based on your suggestions. If you have further concerns or questions, please feel free to share them with us.

---

> > ### Comment · Reviewer_7kvH · 2024-11-26
> >
> > Thanks for your response. Regarding W1: While the manual check and statistical results are helpful, my concern is evaluating how AHP improves CoTs. Specifically, could you analyze the changes in reasoning steps before and after applying AHP, such as the proportion of corrected key reasoning steps? This would better demonstrate the Teacher model's effectiveness.

---

> ### Author Response · Authors · 2024-11-26
>
> Sure. We used the minimum edit distance algorithm (**for convenience, we tokenized by words**) to statistically analyze the proportion of corrupted / corrected key reasoning steps in the corresponding CoTs after applying CCP and AHP. The proportion and histogram data is shown in the table below.
>
> |     | CCP-corrupted key steps | AHP-corrected key steps |
> | --- | --- | --- |
> | average ratio | 0.2089 | 0.2210 |
> | interval distribution: |     |     |
> | 0.0-0.1 | 1731 | 488 |
> | 0.1-0.2 | 697 | 331 |
> | 0.2-0.3 | 444 | 237 |
> | 0.3-0.4 | 325 | 123 |
> | 0.4-0.5 | 191 | 80  |
> | 0.5-0.6 | 125 | 53  |
> | 0.6-0.7 | 73  | 32  |
> | 0.7-0.8 | 67  | 14  |
> | 0.8-0.9 | 42  | 12  |
> | 0.9-1.0 | 110 | 12  |
> | total | 3805 | 1402 |
>
> As we can see, the average proportion of corrected key reasoning steps is approximately 22%, meaning that AHP only needed to modify about 1/5 of the text to correct the CoT. According to interval distribution, we can preliminarily obtain that 75% (488 +331 +237) / 1402) of the data have less than 30% of the modified reasoning steps after applying AHP. This shows the effectiveness of AHP in improving CoT.
>
> * * *
>
> Additionally, to more intuitively show the changes in CoTs before and after applying AHP, we present a case where 22% of the reasoning steps were corrected.
>
> * **task description：** Distinguish deductively valid arguments from formal fallacies.
>
> * **instruction:**
>
>       "Is Fred a cousin of Robert? Is Joe related to Bob? In large families, it is sometimes difficult to keep track of all one's relatives. The following argument seeks to clarify some such relations: First of all, every classmate of Harold who is an ancestor of Frank is not a grandson of Shannon. We may conclude: Every grandson of Shannon is not a classmate of Harold or not an ancestor of Frank."
>       Is the argument, given the explicitly stated premises, deductively valid or invalid?
>       Options:
>       - valid
>       - invalid
>
> * **CoT before applying AHP (original wrong)**
>
>       (1) Every classmate of Harold who is an ancestor of Frank is not a grandson of Shannon: If X = classmate(Harold) AND ancestor(Frank), then X = NOT (grandson(Shannon)).
>
>       Hypothesis: Does it follow that every grandson of Shannon is not a classmate of Harold or not an ancestor of Frank: If X = grandson(Shannon), then X = NOT (classmate(Harold)) OR X = NOT (ancestor(Frank))?
>
>       Let’s see whether the Hypothesis can be deduced from the argument (1) by logical reasoning?
>
>       By (1), we have if X = classmate(Harold) AND ancestor(Frank), then X = NOT (grandson(Shannon)).
>
>       However, the Hypothesis states that if X = grandson(Shannon), then X = NOT (classmate(Harold)) OR X = NOT (ancestor(Frank)).
>
>       The Hypothesis is not a direct consequence of the argument (1). Therefore, the answer is invalid.
>
> * **CoT after applying AHP (corrected)**
>
>       (1) Every classmate of Harold who is an ancestor of Frank is not a grandson of Shannon: If X = classmate(Harold) AND ancestor(Frank), then X = NOT (grandson(Shannon)).
>
>       Hypothesis: Does it follow that every grandson of Shannon is not a classmate of Harold or not an ancestor of Frank: If X = grandson(Shannon), then X = NOT (classmate(Harold)) OR X = NOT (ancestor(Frank))?
>
>       Let’s see whether the Hypothesis can be deduced from the argument (1) by logical reasoning?
>
>       By (1), we have if X = classmate(Harold) AND ancestor(Frank), then X = NOT (grandson(Shannon)).
>
>       By the contrapositive rule in first-order logic, we can rewrite this as: if X = grandson(Shannon), then X = NOT (classmate(Harold)) OR X = NOT (ancestor(Frank)).
>
>       So, it is true that every grandson of Shannon is not a classmate of Harold or not an ancestor of Frank. Therefore, the answer is valid.
>
>
> From these two CoTs, we can see the difference in the text starting from "By (1), we have if X = classmate(Harold) AND ancestor(Frank), then X = NOT (grandson(Shannon)).", which is the point where the reasoning diverges. Prior to this, the reasoning was the same.
>
> In the **corrected CoT**, the step **"By the contrapositive rule in first-order logic, we can rewrite this as: if X = grandson(Shannon), then X = NOT (classmate(Harold)) OR X = NOT (ancestor(Frank))."** applied the contrapositive rule, which is the key reasoning step.
>
> In contrast, the **original wrong CoT** mentions **"However, the Hypothesis states that if X = grandson(Shannon), then X = NOT (classmate(Harold)) OR X = NOT (ancestor(Frank)). The Hypothesis is not a direct consequence of the argument (1)."** Here, it failed to recognize the contrapositive rule and incorrectly concluded that the hypothesis could not be directly deduced from the premise.
>
> From this point onward, the reasoning path and final conclusion differ. We hope that the analysis above demonstrates the effectiveness of the Teacher model in improving CoTs under the AHP method.

---

### Official Review · Reviewer_9WvQ · 2024-10-31

**Soundness:** 3
**Presentation:** 2
**Contribution:** 2
**Rating:** 5
**Confidence:** 4

**Summary:**

The paper introduces EDIT (mistakE-Driven key reasonIng step distillaTion), a method aimed at enhancing the reasoning ability of smaller language models (SLMs) by distilling key reasoning steps from large language models (LLMs). Unlike traditional approaches that focus solely on correct answers, EDIT leverages dual chains of thoughts—pairs of reasoning processes that share a similar path but reach different conclusions—to identify critical reasoning steps using a minimum edit distance algorithm. This technique improves the SLM's ability to capture essential reasoning steps, thereby increasing the accuracy of its conclusions, as demonstrated through experiments on both in-domain and out-of-domain reasoning tasks.

**Strengths:**

1.	The introduction of dual chains of thoughts (CoTs) and the application of a minimum edit distance algorithm to identify crucial reasoning steps represent a unique approach to improving the reasoning capabilities of smaller language models (SLMs).
2.	The method used in this work can improve the interpretability and generalization of reasoning abilities in smaller language models.
3.	The methodology is sound, incorporating the generation of corrected CoTs and corrupted originally correct CoTs to create a rich training set for the SLMs. The results indicate that EDIT outperforms simple fine-tuning techniques.

**Weaknesses:**

1.	The paper’s presentation could be enhanced by offering a clearer explanation of the mistake-driven key reasoning step distillation process shown in Figure 2. Readers may find it challenging to understand the purpose of obtaining the corrupted CoT and rectified CoT from the figure alone, so providing a more explicit explanation at the beginning of the methodology section is recommended.
2.	In the process of "Rectifying Wrong CoT," does providing the correct answer to the large model guarantee that it will generate a completely correct output? The quality of the data generated through this part of the method lacks further discussion. Additionally, how is it ensured that the rectified CoT retains similar reasoning steps, and does this impact the calculation of the minimum edit distance in later stages?
3.	The cases lack a comparison between the reasoning process generated by Edit and that of the ground truth. It remains unclear whether the reasoning process generated by Edit can truly capture the key steps in the ground truth.

**Questions:**

1.	How are the weight values of α and β set, and how do they impact the performance of KRSL?
2.	Are there metrics to assess whether the reasoning steps generated by EDIT are more accurate compared to the original SFT?

---

> ### Author Response · Authors · 2024-11-22
> **Response of Weaknesses**
>
> Thank you for your thorough and insightful comments. We greatly appreciate your valuable feedback and the effort you have invested in reviewing our work. Below, we address your concerns point by point.
>
> > **W1:** The paper’s presentation could be enhanced by offering a clearer explanation of the mistake-driven key reasoning step distillation process shown in Figure 2.
>
> **A1:** Thank you for your suggestion. We will provide a more explicit explanation of Figure 2 at the beginning of the methodology section to enhance the clarity and accessibility of our proposed approach.
>
> > **W2:** In the process of "Rectifying Wrong CoT," does providing the correct answer to the large model guarantee that it will generate a completely correct output? The quality of the data generated through this part of the method lacks further discussion. Additionally, how is it ensured that the rectified CoT retains similar reasoning steps, and does this impact the calculation of the minimum edit distance in later stages?
>
> **A2:**
>
> * **On data quality and reasoning consistency:** Yes, the answer hint prompt can ensure the LLM generates a correct response. In our manual random inspection of 100 samples (50 rectified CoTs and 50 corrupted CoTs), only 7 rectified CoTs showed logical inconsistencies between the reasoning process and the final answer. Leveraging the teacher LLM's strong contextual understanding and autoregressive nature, the generated CoTs exhibit similar reasoning processes but with different conclusions, while maintaining logical consistency. We appreciate your suggestion and will include an evaluation of data quality in the revised paper.
>
> * **On ensuring reasoning similarity:** As detailed in Section 3.2, we carefully designed prompts to guarantee the synthesis of dual CoTs with desired properties. Both the Answer Hint Prompt (AHP) and Contrastive CoTs Prompt (CCP) share the same contextual examples with the CoTs Extraction Prompt (CEP), except that AHP introduces a correct answer hint while retaining identical questions and CoTs. This ensures that the LLM generates rectified CoTs with similar reasoning steps. For CCP, the contextual examples’ CoTs adhere to specific properties where the reasoning processes are similar but lead to different conclusions. These are sampled from the constructed dual CoTs ($D^{-}\_{dual}$ and $D^{−+}\_{dual}$​), with the teacher model performing a continuation task to synthesize negative CoTs for downstream queries.Through this design, we ensure the desired data properties without compromising the subsequent minimum edit distance calculation.
>
>
> > **W3:** The cases lack a comparison between the reasoning process generated by Edit and that of the ground truth. It remains unclear whether the reasoning process generated by Edit can truly capture the key steps in the ground truth.
>
> **A3:** We apologize for not presenting this comparison more clearly in the main text.
>
> From Tables 19 and 20, we observe that both the teacher and Std-CoT models make mistakes at the same positions in their reasoning processes, even though the nature of their mistakes differs. These positions can be considered key reasoning steps. In contrast, the EDIT CoT demonstrates correct reasoning at these corresponding positions (highlighted in green), which subsequently leads to a correct conclusion.
>
> Additionally, we noticed an interesting phenomenon not observed in every case. For example, in Table 20, while the Std-CoT and teacher models both adopt a logic of enumerating and analyzing each option, EDIT raises **issues or questions** for each option and then answers them. This suggests that EDIT, through learning key reasoning steps, avoids overfitting to the teacher CoT’s reasoning steps and instead adapts its reasoning logic to solve the problem effectively.

---

> ### Author Response · Authors · 2024-11-22
> **Response of Questions**
>
> > **Q1:** How are the weight values of α and β set, and how do they impact the performance of KRSL?
>
> **A4:** As described in Line 310, the values of α and β were determined empirically through grid search. We observed that learning from positive samples has a more significant impact than learning from negative samples. Furthermore, the performance of the final model is sensitive to the relative weighting of the two loss terms.
>
> To explore this further, we varied the values of α and β, as shown in the table below:
>
> | Group | α   | β   | BBH-test | BB-sub | AGIEval | ARC-E | ARC-C | AVG |
> | --- | --- | --- | --- | --- | --- | --- | --- | --- |
> | A   | 0   | 0   | 59.7 | 30.0 | 24.5 | 61.9 | 45.5 | 44.32 |
> | B   | 0   | 0.025 | 59.0 | 30.2 | 24.1 | 62.1 | 45.9 | 44.26 |
> | C   | 1   | 0   | 60.2 | 30.5 | 23.4 | 62.7 | 48.0 | 44.96 |
> | D   | 1   | 0.025 | **60.9** | **31.1** | **25.9** | **64.1** | **50.5** | **46.50** |
> | E   | 1   | 0.05 | 59.7 | 30.0 | 24.7 | 61.9 | 45.5 | 44.36 |
>
> Increasing α from 0 to 1 (comparing A to C or B to D) results in significant performance gains across most benchmarks. However, further increasing β beyond 0.025 leads to a noticeable drop in performance, indicating that the two loss terms in eq.(6) need to strike a balance to achieve optimal performance. Excessive dominance of either term negatively impacts model training. Thus, the two terms exhibit a collaborative yet adversarial relationship.
>
> > **Q2:** Are there metrics to assess whether the reasoning steps generated by EDIT are more accurate compared to the original SFT?
>
> **A5:** In Section 5.2, we evaluated the reasoning quality of CoTs generated by the teacher, Std-CoT, and EDIT from the perspective of key reasoning steps, with GPT-4 serving as the judge. The results (Figure 5, right) show that EDIT produces CoTs whose score distributions are closer to those of the teacher model than Std-CoT, indicating that EDIT effectively learns to generate key reasoning steps beneficial for solving the task.
>
> We greatly value your thoughtful feedback and are committed to improving our work based on your suggestions. If you have further concerns or questions, please feel free to share them with us.

---

### Official Review · Reviewer_BT6K · 2024-11-01

**Soundness:** 3
**Presentation:** 3
**Contribution:** 3
**Rating:** 6
**Confidence:** 4

**Summary:**

This paper focuses on distillation methods for SLMs through learning key reasoning steps from dual CoTs data. Specifically, they first collect CoTs data annotated by teacher LLMs including both the correct and wrong ones, and then generate their dual CoTs data, which possess similar reasoning paths but divergent conclusions. On this basis, they identify the key reasoning step via the minimum edit distance and employ a fine-grained loss for finetuning, thereby allowing students to learn key reasoning steps instead of simply imitating teachers’ reasoning forms. Extensive experiments demonstrate their effectiveness and superiority.

**Strengths:**

1.This work designs dual CoTs data by prompting teacher LLMs, which can be utilized to  identify key reasoning steps through the minimum edit distance algorithm.

2.This work proposes a fine-grained loss based on the dual CoTs data, which enables students to focus more on key reasoning steps and reduce errors, rather than simply imitating teachers’ reasoning forms, further improving reasoning.

3.The proposed EDIT outperforms baselines on both in-domain and out-of-domain benchmarks, demonstrating its effectiveness and versatility. Further analysis also shows that EDIT can generate high-quality CoTs with more correct key reasoning step, and logical errors in dual CoTs benefit more for student SLMs.

**Weaknesses:**

1. In L312, why is greedy decoding used for text generation in inference? Is it used to ensure the similarity between dual CoTs data? If not, how can we guarantee the formal similarity of dual data, that is, only the key steps are different, while the reasoning steps, forms, and even words of other parts remain consistent? And will there be a situation where non-key reasoning steps are identified as key steps?

2. It seems that the rectified teacher’s wrong CoTs contributes more in EDIT, as in Table 1, EDIT w/o KRSL outperforms w/o RWC, and in Table 2, D_dual^- outperforms D_dual^+. Therefore, what if not using RWC in the first step and only using D_dual^ + in the second step?

3. To better show the superiority of the method, can you please compare EDIT with some of the latest baseline methods?

4. Typo: Table 24 and Table 25 are reversed.

**Questions:**

See weakness.

---

> ### Author Response · Authors · 2024-11-22
> **Responses of W1-W2**
>
> Thank you for your thorough and constructive comments. We sincerely appreciate the time and effort you have put into reviewing our work. Below, we address your concerns point by point.
>
> > **W1:** In L312, why is greedy decoding used for text generation in inference? Is it used to ensure the similarity between dual CoTs data?  How can we guarantee the formal similarity of dual data, that is, only the key steps are different? And will there be a situation where non-key reasoning steps are identified as key steps?
>
> **A1:**
>
> * In L312, we set the student model to use greedy decoding during the evaluation phase for text generation, while the teacher generates data using sampling-based methods (parameters detailed in Table 15).
>
> * In Section 3.2, we describe how carefully designed prompts ensure the synthesis of dual CoTs with specific desired properties. Leveraging the teacher LLM's strong contextual learning abilities, we crafted three key prompts:
>
>   * **Answer Hint Prompt (AHP):** Builds upon the CoT Extraction Prompt (CEP) by adding correct-answer hints to examples, while keeping the CoT reasoning process unchanged.
>   * **Contrastive CoTs Prompt (CCP):** Incorporates paired CoTs (sampled from \$D^{-}\_{dual}\$ and \$D^{-+}\_{dual}\$) with differing conclusions but similar reasoning structures to teach dual CoTs' contrasting characteristics. The teacher model generates negative CoTs via a continuation task in this context.
> * Based on a manual review of 100 randomly sampled examples, almost no instances were found where non-key reasoning steps were incorrectly identified as key steps. The teacher LLM’s strong autoregressive reasoning ensures that generated CoTs are logically consistent yet differ in key reasoning steps as intended.
>
>
> > **W2:** It seems that the rectified teacher’s wrong CoTs contributes more in EDIT, as in Table 1, EDIT w/o KRSL outperforms w/o RWC, and in Table 2, D_dual^- outperforms D_dual^+. Therefore, what if not using RWC in the first step and only using D_dual^ + in the second step?
>
> **A2:** We conducted experiments to analyze the impact of this setup. Below are the methods used for comparison:
>
> * A：baseline method using standard CoT distillation (Std-CoT).
>
> * B：excludes RWC in the first step and only uses $D^{+}_{dual}$ in the second step
>
> * C：Excludes RWC in the first step and uses all dual datasets in the second step
>
> * D: uses RWC in the first step and skips second step
>
> * E: Our complete method (EDIT)
>
>   **Results:**
>
>   1. **Impact of the second step:**
>
>     * Comparing **A** vs. **B** shows the second step improves performance.
>     * Comparing **B** vs. **C** demonstrates that including $D^{-}_{dual}$​ further enhances performance.
>   2. **Impact of the first step:**
>
>     * Comparing **A** vs. **D** shows that RWC significantly improves in-domain task performance.
>     * Comparing **A** vs. **C** shows that KRSL improves out-of-domain task performance.
>   3. **Combined effect:**
>
>     * Comparing **A** vs. **E**, the combination of RWC and KRSL yields significant improvements in both in-domain and out-of-domain tasks.
>
>   | Group | Method | BBH-test | BB-sub | AGIEval | ARC-E | ARC-C | AVG |
>   | --- | --- | --- | --- | --- | --- | --- | --- |
>   |     | **In-domain** | √   | ×   | ×   | ×   | ×   |     |
>   | **A** | w/o RWC + w/o KRSL | 54.2 | 28.7 | 21.6 | 59.6 | 45.1 | 41.80 |
>   | **B** | w/o RWC + KRSL on \$D^{+}_{dual}\$​ | 55.1 | 30.1 | 24.1 | 60.3 | 44.1 | 42.70 |
>   | **C** | w/o RWC + KRSL on \$D^{+}_{dual}\cup D^{-}\_{dual}\$ | 55.4 | 30.1 | 24.2 | 63.6 | 48.3 | 44.30 |
>   | **D** | w/ RWC + w/o KRSL | 59.7 | 30.0 | 24.5 | 61.9 | 45.5 | 44.30 |
>   | **E** | w/ RWC + w/ KRSL on \$D^{+}_{dual}\cup D^{-}\_{dual}\$ | **60.9** | **31.1** | **25.9** | **64.1** | **50.5** | **46.50** |

---

> ### Author Response · Authors · 2024-11-22
> **Responses of W3-W4**
>
> > **W3:** To better show the superiority of the method, can you please compare EDIT with some of the latest baseline methods?
>
> **A3:** Sure, we compared our method against recent approaches [1, 2]. Hsieh et al. (2023) [1] proposed a multi-task CoTs distillation method that distills rationales and answers separately. Chen et al. (2024) [2] proposed to learn the mutual relationship of the two tasks from an Information Bottleneck perspective.
>
> From the experimental results in the table below, we observe that the EDIT method outperforms these baselines on three benchmarks, particularly for in-domain tasks. Although these methods excel in out-of-domain tasks, it is important to note that their evaluation only requires generating the answer. Since the learning objectives of these methods include directly fitting to the answer labels, this approach often results in low logical consistency between the rationale and answer generated by the student model. In contrast, our method is an improvement based on Std-CoT, where the reasoning generation mode ensures that the rationale and answer are coherent, thereby guaranteeing the fidelity of reasoning.
>
> | Method | BBH-test | BB-sub | AGIEval | ARC-E | ARC-C | AVG |
> | --- | --- | --- | --- | --- | --- | --- |
> | In-domain | √   | ×   | ×   | ×   | ×   |     |
> | MT-CoT | 56.8 | 30.3 | 22.0 | 49.4 | 38.2 | 39.3 |
> | SCOTT | 42.4 | 18.8 | 13.0 | 45.7 | 34.1 | 30.8 |
> | Std-CoT | 54.2 | 28.7 | 21.6 | 59.6 | 45.1 | 41.8 |
> | Hsieh et al. (2023)[1] | 42.4 | 27.7 | 28.8 | **68.5** | 48.6 | 43.2 |
> | Chen et al. (2024)[2] | 42.9 | 24.3 | **29.2** | 68.4 | 49.3 | 42.8 |
> | EDIT(ours) | **60.9** | **31.1** | 25.9 | 64.1 | **50.5** | **46.5** |
>
> [1] Hsieh C Y, Li C L, Yeh C, et al. Distilling Step-by-Step! Outperforming Larger Language Models with Less Training Data and Smaller Model Sizes[C]//Findings of the Association for Computational Linguistics: ACL 2023. 2023: 8003-8017.
>
> [2] Chen X, Huang H, Gao Y, et al. Learning to Maximize Mutual Information for Chain-of-Thought Distillation[C]//Findings of the Association for Computational Linguistics: ACL 2024. 2024: 6857-6868.
>
> > **W4:** Typo: Table 24 and Table 25 are reversed.
>
> **A4:** Thank you for pointing this out. We will correct this typo and thoroughly review the manuscript for any additional issues.
>
> We greatly value your thoughtful feedback and are committed to improving our work based on your suggestions. If you have further concerns or questions, please feel free to share them with us.

---

### Official Review · Reviewer_6mQT · 2024-11-04

**Soundness:** 2
**Presentation:** 2
**Contribution:** 2
**Rating:** 3
**Confidence:** 4

**Summary:**

This paper introduces a novel distillation method based on dual Chain-of-Thought (CoT) data that share similar reasoning paths but reach different conclusions. The approach begins by retaining all CoT data generated by the teacher model (GPT-3.5-turbo), regardless of correctness. The authors then design two prompts to guide the teacher in producing dual CoTs, ensuring they follow similar intermediate reasoning steps but arrive at divergent conclusions. Using a minimum edit distance algorithm, they identify the critical reasoning steps in these dual CoTs and apply a fine-grained loss function to optimize the likelihood of these key steps.

**Strengths:**

1. Distilling the strong reasoning capabilities of a large language model (LLM) into a smaller LLM is an intriguing and valuable research topic.

2. The analysis of dual CoTs with different error types provides a more comprehensive approach to the distillation process, offering insights that could guide further refinement of Chain-of-Thought distillation methods.

**Weaknesses:**

1. The novelty of the approach is limited, as many existing works already utilize mistake retrieval to provide tailored guidance and improve model performance during Chain-of-Thought (CoT) inference.

2. The method relies on constructing dual CoT datasets composed of positive-negative CoT pairs with similar intermediate reasoning steps but divergent conclusions. In practice, however, such pairs are often scarce, and the annotation process can be costly and labor-intensive, typically requiring manual selection. This could be a significant limitation unless the generation prompt achieves a high level of accuracy without human oversight.

3. The performance improvement over the teacher model is minimal, suggesting that the distillation process achieves only marginal gains and has room for further enhancement.

4. The method's reliance on teacher model errors could introduce inconsistencies, particularly if certain error types are more prevalent or systematically biased in the teacher model. Also, by focusing intensively on key reasoning steps, there may be a risk of overfitting certain reasoning patterns, potentially impacting generalization in scenarios that require varied reasoning styles.

**Questions:**

1. How about distilling the strong reasoning capabilities of a large language model (LLM) into a smaller LM? Let's say a 1.5B model.

2. Why is there a huge gap in performance between the teacher model and the student model?

3. What is the scalability of EDIT? What if there are not enough "high-quality" positive-negative CoT pairs that follow similar intermediate reasoning steps but lead to divergent conclusions?

---

> ### Author Response · Authors · 2024-11-22
>
> Thank you for your thorough and constructive comments. We sincerely appreciate the time and effort you have put into reviewing our work. Below, we address your concerns point by point:
>
> > **W1:** The novelty of the approach is limited...
>
> **A1:** We have discussed related works on learning from errors in Lines 129-140. Existing methods often rely on relatively shallow approaches, such as directly adding "counterfactual reasoning" markers to erroneous data for training (Wang et al., 2023a), utilizing different LLMs to correct errors (An et al., 2023), or embedding guidance for error retrieval in prompts (Sun et al., 2024).In contrast, our work adopts a novel perspective by leveraging the comparison between erroneous and correct data to identify critical reasoning steps, thereby enabling more precise optimization.
>
> > **W2:** Dual CoT pairs are scarce, and their creation is costly and labor-intensive, often requiring manual selection unless prompts achieve high accuracy autonomously.
>
> **A2:** We propose a method to create high-quality dual CoT pairs, detailed in Section 3.2. Inspired by the strong contextual learning abilities of large models, we designed three prompts: Answer Hint Prompt (AHP), Contrastive CoTs Prompt (CCP), and CoTs Extraction Prompt (CEP), all sharing identical contextual examples.The difference lies in the additional correct answer hints introduced in AHP compared to CEP, while CCP’s examples adhere to specific data properties: reasoning processes are similar but lead to different conclusions. These are sampled from constructed dual CoT pairs ($D^{-}\_{dual}$​ and $D^{-+}\_{dual}$​), with a teacher model performing continuation tasks to generate negative CoTs for downstream queries.We randomly sampled 100 dual CoT pairs and verified their adherence to the expected data properties, ensuring logical consistency between reasoning processes and conclusions. Based on these results, we finalized the design of the two prompts to ensure high accuracy.
>
> > **Q1:** How about distilling the strong reasoning capabilities of a large language model (LLM) into a smaller LM? Let's say a 1.5B model.
>
> **A3:** In our work, we have experimented with distillation on a smaller LM (TinyLLaMA-1.1B), as mentioned in Line 373. The results, shown in Figures 4 and 6, demonstrate significant improvements on in-domain benchmarks and surpass baseline performance on three out-of-domain benchmarks.
>
> > **W3&Q2:** The performance improvement over the teacher model is minimal. Why is there a huge gap in performance between the teacher model and the student model?
>
> **A4:** Many open-source LLM projects synthesize a substantial amount of high-quality instruction data (>100k), enabling student models to outperform teacher models. In contrast, our training dataset contains only 5,207 samples, which likely contributes to the performance gap between the student and teacher models.Unlike recent works focusing on the Instruction Following paradigm and improving synthesized data quality, our research emphasizes advancements in the distillation paradigm. We believe this paradigm improvement offers greater potential contributions to this field compared to data quality enhancements.
>
> > **Q3:** What is the scalability of EDIT? What if there are not enough "high-quality" positive-negative CoT pairs?
>
> **A5:** EDIT's effectiveness depends on the quality of the synthesized dual CoTs. If their textual differences do not reflect true core reasoning differences—e.g., containing irrelevant reasoning steps or stop words—EDIT's impact would be diminished.Our method is designed to guide smaller models to focus on key reasoning steps in long reasoning chains, avoiding overfitting to easier tokens. Therefore, we carefully designed prompts to ensure the synthesized data possesses the specific properties required to ground critical reasoning steps. Without this foundation, the effectiveness of our approach would be limited.
>
> We greatly value your thoughtful feedback and are committed to improving our work based on your suggestions. If you have further concerns or questions, please feel free to share them with us.

---

### Note · Authors · 2024-12-16

I have read and agree with the venue's withdrawal policy on behalf of myself and my co-authors.